# Alpibectir–Ethionamide combination (AlpE) for the treatment of tuberculosis

Zainab Edoo[1,2,14], Camille Grosse[1,3,14], Thomas Maitre[4,5,14], Rosangela Frita[1,2,14], Aurélie Chauffour [4], Laure Fournier Le Ray[4], Alexandre Godmer [4,6], Alexandra Aubry[4,7], Marilyne Bourotte[2], Rudy Antoine [1], Lina Tawk[1], Stéphanie Slupek[1], Vincent Trebosc [8], Birgit Schellhorn[8], Aurore Dreneau[2], Line Hofmann[2], Christian Kemmer[8], Sergio Lociuro[2,8], Glenn E. Dale[2,8], Françoise Jung [8], Esther Pérez-Herrán [9], Alfonso Mendoza [9], Maria Jose Rebollo López[9], Sonja Ghidelli-Disse[10], Thilo Werner[10], Lluis Ballell[9], David Barros-Aguirre [9], Vanessa Mathys [11], Karine Soetaert[11], Véronique Megalizzi[3], René Wintjens [3], Marc Gitzinger[2,8], Benoit Deprez [12,13,15], Nicolas Veziris[4,6], Modesto J. Remuiñán[9,15], Nicolas Willand[12,15] ✉, Michel Pieren[8,15] ✉ & Alain R. Baulard [1,13,15] ✉

Ethionamide (Eto) and prothionamide (Pto) are second-line antibiotics used for tuberculosis (TB) treatment. Both are prodrugs whose antibacterial activity depends on bioactivation by oxidases in *Mycobacterium tuberculosis*, including the Baeyer-Villiger monooxygenase MymA. Through biophysical, genetic, and cellular assays, we show that the clinical candidate alpibectir (Alp, BVL-GSK098) binds the transcriptional regulator VirS, increasing MymA expression and potentiating Eto and Pto activity. Alpibectir also boosts the activity of the corresponding host-derived sulfoxide metabolites. We additionally show that alpibectir exhibits intrinsic antibacterial activity via overexpression of the *mymA* operon. The alpibectir/Eto (AlpE) combination is rapidly bactericidal in vitro and in mice, lowers the frequency of spontaneous resistance of Eto, and remains active on Eto- and isoniazid-resistant strains, including isolates with *inhA* promoter mutations. Alpibectir was safe in a Phase 1 human clinical trial. Together with the potentiation data presented here, these findings highlight its potential to optimize TB chemotherapy by reducing Eto/Pto doses, which can minimize dose-related side effects, enhancing adherence.

Tuberculosis (TB) is one of the leading causes of mortality worldwide[1]. Its causative agent is the bacterial pathogen *Mycobacterium tuberculosis* (*M. tuberculosis*). The impact of the COVID-19 pandemic has been profound, resulting in a setback in diagnosis and treatment, reversing years of hard-won progress in reducing TB fatalities and underscoring the critical need to strengthen research and medical efforts to combat this persistent scourge. The growing threat of drug-resistant TB exacerbates this challenge, necessitating the use of less efficient and less safe second-line drugs[2,3], thereby prolonging the treatment duration and diminishing its success.

Thioamides, including ethionamide (Eto) and prothionamide (Pto), are recommended by the World Health Organization (WHO) as second-line agents for treating drug-resistant pulmonary TB[3] and as first-line treatment for TB meningitis[4]. Despite their efficacy, these drugs pose a hurdle to treatment adherence due to dose-dependent adverse events[5,6]. Eto and Pto are prodrugs, so their antibacterial

activity is linked to their level of bioactivation. The main bioactivating enzyme is the mycobacterial monooxygenase EthA, which initiates a series of catalytic events leading to the formation of an Eto/Pto-NAD adduct targeting InhA, an enoyl-reductase playing a pivotal role in the synthesis of mycolic acid in the mycobacterial cell envelope[7,8]. The expression of EthA is repressed by the transcriptional regulator EthR[9]. Inhibition of EthR by small molecules was shown to enhance EthA expression[10–17]. The most potent EthR inhibitor, BDM41906, demonstrated increased biotransformation of Eto and its major metabolite ethionamide-sulfoxide (Eto-SO) by EthA, and increased the susceptibility of *M. tuberculosis* to both Eto and Eto-SO in vitro[18] and to Eto in vivo[13,18,19]. Although this was an exciting proof of concept, the therapeutic potential was constrained. The continuous use of Eto and Pto over the last 60 years has led to the selection of resistant clinical isolates mutated in *ethA*, rendering these strains resistant to Eto/Pto, even in combination with BDM41906.

An unexpected rebound of optimism in this strategy emerged with our discovery of compounds capable of stimulating the expression of alternative Eto/Pto activation pathways[20,21]. Specifically, SMARt751 was shown to interact with VirS and induce the overexpression of the *mymA* operon, which was concomitantly identified for its ability to activate Eto. Importantly, by stimulating Eto activation through the upregulation of MymA, SMARt751 not only enhanced Eto efficacy but also circumvented resistance to Eto in clinical strains mutated in *ethA*[21]. However, despite these promising effects, SMARt751 also triggered activation of peroxisome proliferator-activated receptors alpha (PPARα) and the constitutive androstane receptor (CAR) at all tested doses in vivo. Due to these potential drug metabolism and pharmacokinetics (DMPK) and safety concerns, further development of SMARt751 was not pursued. Nevertheless, a multi-parametric lead optimization program led to the design of alpibectir, which demonstrated microbiological efficacy, pharmacokinetic properties, and safety profiles adequate for clinical development (manuscript in preparation).

Here, we elucidated the complex mode of action of alpibectir, linked to its molecular interaction with VirS. The ability of alpibectir to stimulate the *mymA* operon leads to a triple effect on the bacteria. First, overexpression of *mymA* improves the bioactivation of both Eto and its major metabolite Eto-SO. Importantly, alpibectir overcomes acquired Eto resistance by circumventing EthA-dependent Eto bioactivation. Finally, alpibectir has its own growth inhibitory effect, intriguingly, also subordinated to the overexpression of the *mymA* operon. The confluence of these effects allows alpibectir to potentiate the antibacterial activity of Eto in vivo in a 4-week acute murine model of TB and makes the combination a powerful anti-TB medication candidate. Alpibectir was safe and well tolerated in a Phase I clinical trial[22] and a Phase 2a study (NCT05473195) has established a proof of concept for the combination in patients with pulmonary TB[23]. Alpibectir is envisaged for future use in combination with Eto and Pto in patients with isoniazid (INH) mono-resistant or multidrug-resistant (MDR) TB, including TB meningitis.

## Results

### Alpibectir enhances Eto activity against *M. tuberculosis*
The antibacterial activity of the alpibectir/Eto (AlpE) combination on *M. tuberculosis* was first assessed using a checkerboard microtiter assay (Fig. 1). Bacterial growth was monitored by quantifying fluorescence intensity of a GFP-expressing *M. tuberculosis* H37Rv strain after 5 days of incubation. Concentration-response curves of Eto were plotted for each individual concentration of alpibectir tested. $IC_{50}$ and $IC_{90}$ extrapolated from each curve refer to the concentrations of Eto required to inhibit 50% and 90% of bacterial fluorescence, respectively. The $IC_{50}$ and $IC_{90}$ values of Eto alone were found to be 1.0 mg/L and 3.7 mg/L. In this assay, a concentration of 0.03 mg/L (100 nM) of alpibectir was found to reduce the $IC_{90}$ value of Eto by more than 100-fold

(down to 0.03 mg/L). Maximum potentiation of Eto activity (almost 200-fold reduction in the $IC_{90}$ of Eto) was observed at a concentration of 0.1 mg/L (300 nM) of alpibectir (Fig. 1E).

Strikingly, we also observed concentration-dependent growth inhibition by alpibectir in the absence of Eto (Fig. 1C). We confirmed through optical density measurements that the decrease in fluorescence observed upon treatment with alpibectir alone was due to bacterial growth inhibition and not to dose-dependent quenching of bacterial fluorescence by alpibectir (SI Fig. 1). These results reveal that alpibectir not only potentiates Eto but can also contribute by itself to the inhibition of *M. tuberculosis* growth. Interestingly, close structural analogues of alpibectir that lack Eto-boosting activity also lack intrinsic growth inhibition, consistent with a narrow structural requirement shared between both activities (SI Fig. 2).

### Alpibectir boosts Eto activity in infected macrophages
In the context of infection, *M. tuberculosis* primarily resides within host cells[24]. Therefore, penetration of host cells and the effectiveness of anti-TB drugs within the intracellular environment are crucial for treatment success. The antibacterial activity of Eto in the presence of alpibectir was assessed on luminescent *M. tuberculosis* H37Rv in infected THP-1 cells by measuring the luminescence of intracellular bacteria. As a control, macrophages were confirmed to be viable irrespective of drug treatment (SI Fig. 3). The maximum intrinsic activity and boosting effect of alpibectir were observed at a concentration of ≤ 0.03 mg/L, reducing the $IC_{90}$ of Eto from 0.1 mg/L (in the absence of alpibectir) to 0.01 mg/L (Fig. 1F).

These experiments demonstrate that alpibectir is highly potent in increasing the antibacterial activity of Eto in infected macrophages. A comparable boost of Eto activity was observed in an independent macrophage survival assay that quantifies protection from infection-induced macrophage lysis (SI Fig. 4). In this assay, however, alpibectir used alone showed no detectable intrinsic activity, suggesting that while it can inhibit intracellular bacterial growth, the effect is insufficient on its own to prevent macrophage lysis (Fig. 1F).

### Alpibectir preferentially activates the VirS/MymA pathway
We then assessed the impact of alpibectir on the transcriptome and proteome of *M. tuberculosis*. When *M. tuberculosis* was exposed to a concentration of 0.1 mg/L of alpibectir, RNA-seq identified 37 genes that were upregulated by more than two-fold (Fig. 2). Notably, the most prominently overexpressed mRNAs corresponded to the *virS-mymA* regulon (*rv3082c* to *rv3089*), a profile reminiscent of what was previously observed with SMARt751[21]. The transcription of *rv0077c*, the gene encoding the oxidoreductase EthA2[21], was also increased, although to a lesser extent. The addition of Eto did not change the transcriptomics profile induced by alpibectir, indicating that Eto does not interfere with the alpibectir-mediated induction of the *mymA* operon (SI Table 1).

Next, we quantified modifications of the *M. tuberculosis* global proteome upon treatment with alpibectir (SI Table 1). We compared the total protein content of an *M. tuberculosis* culture treated with alpibectir to that of a culture treated with DMSO using isobaric labelling-based mass spectrometry[25]. Out of the approximately 2,200 proteins identified, 9 proteins were overproduced upon alpibectir treatment. Eight of them correspond to the proteins of the *virS-mymA* regulon and the ninth is the EthA2 protein.

These findings confirm that the observed rise in transcription of the *virS-mymA* and *ethR2-ethA2* regulons induced by alpibectir treatment leads to an increased production of their corresponding proteins. Altogether, this underscores the exceptional functional selectivity of alpibectir, which has virtually no other effect than predominantly stimulating the *mymA* operon, and to a lesser extent the *ethA2* gene. Notably, while a tenfold reduction in EthA protein

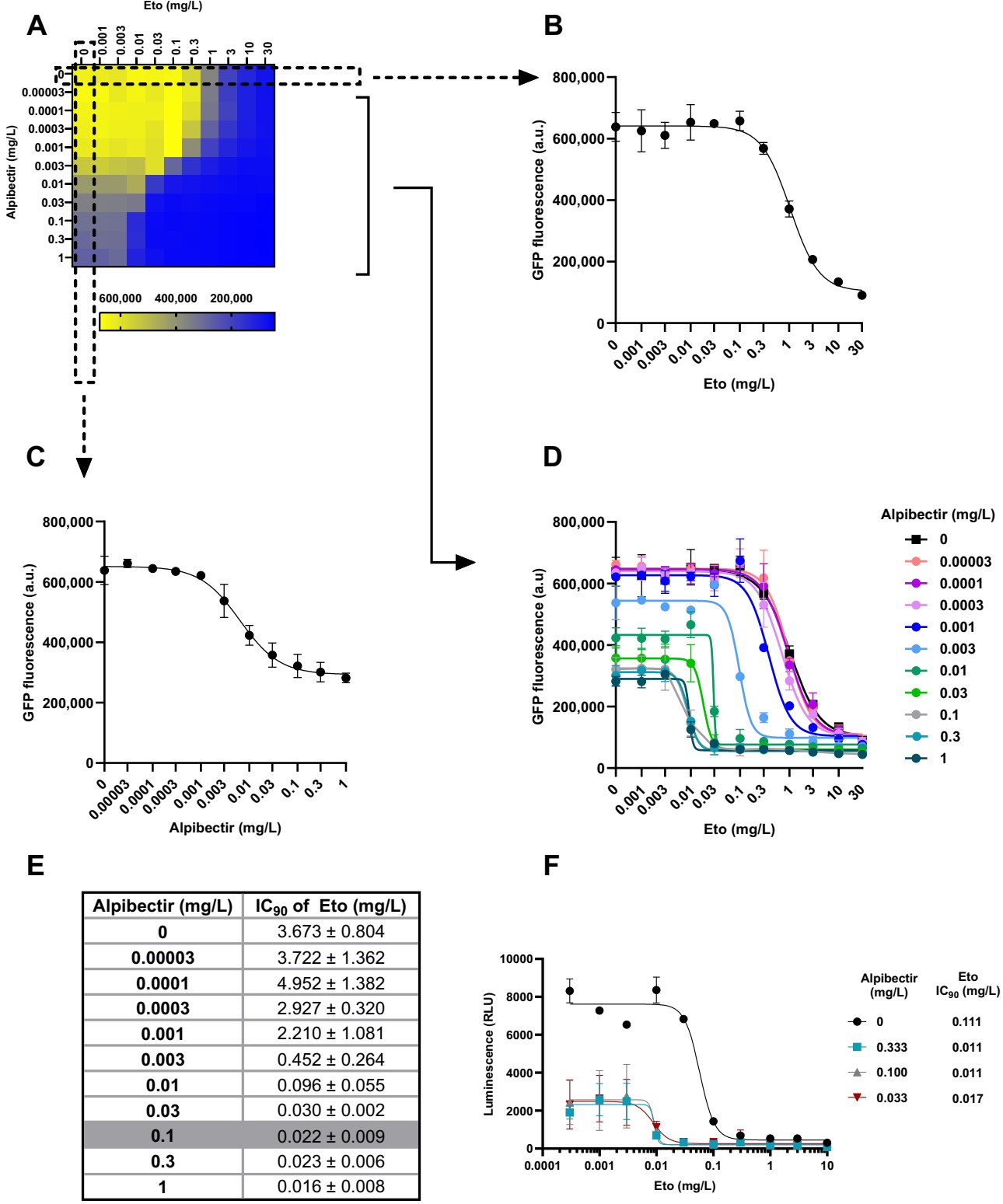

**Fig. 1 | Effect of alpibectir on the susceptibility of *M. tuberculosis* H37Rv to Eto.**
**A** Checkerboard assay with increasing Eto and alpibectir concentrations. The fluorescence of a GFP-expressing *M. tuberculosis* strain was measured after 5 days of incubation with the corresponding drug concentrations. **B** Concentration-response curve obtained for Eto in the absence of alpibectir. **C** Concentration-response curve obtained for alpibectir in the absence of Eto. **D** Eto concentration-response curves for each alpibectir concentration. **E** IC$_{90}$ of Eto at different concentrations of alpibectir. **F** Eto concentration-response curves and IC$_{90}$ of Eto alone or in combination with alpibectir against intracellular H37Rv. Luminescence of intracellular H37Rv was measured after 5 days of incubation. Results shown are representative of 3 biological replicates. Data points represent the mean ± standard deviation of 2 technical replicates. Source data are provided as a Source Data file.

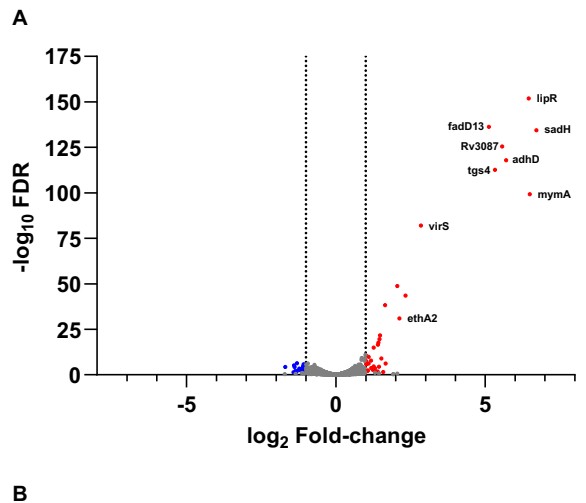

**A**

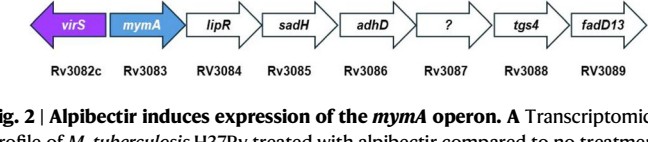

**B**

Rv3082c — Rv3083 — RV3084 — Rv3085 — Rv3086 — Rv3087 — Rv3088 — RV3089

**Fig. 2 | Alpibectir induces expression of the *mymA* operon. A** Transcriptomics profile of *M. tuberculosis* H37Rv treated with alpibectir compared to no treatment. The volcano plot shows the $\log_2$ fold-change (L2FC) values and false discovery rates (FDR) for each gene. Dotted lines represent the cut-off value of |L2FC| > 1. Significantly overexpressed and underexpressed genes ($-\log_{10}$ FDR > 1.3) are shown in red and blue, respectively. Results are shown for 3 biological replicates. **B** Genetic organization of the *virS-mymA* regulon. In the absence of alpibectir, the expression of the regulon is uniformly low. Treatment with alpibectir induces the expression of *virS*, as well as of the entire *mymA* operon, which comprises 7 genes. *virS*, purple; *mymA*, blue. RNAseq revealed a monocistronic expression with no apparent transcriptional relay inside the operon, suggesting that the promoter/operator upstream of *mymA* is the only one initiating and controlling the expression of the entire set of genes of the operon.

production was observed, no significant change in *ethA* expression (less than a 2-fold difference) was detected.

## Alpibectir modifies VirS activity through direct interactions

Transcriptomic data suggest that alpibectir may interact with the transcriptional regulator VirS to stimulate the expression of the *virS/mymA* operon. Here we wanted to elucidate whether this stimulation involves direct or indirect interactions between alpibectir and VirS. First, we developed a synthetic gene circuit in mammalian cells that quantitatively reports the expression of the secreted alkaline phosphatase (SEAP) gene, designed to sense interactions between the bacterial protein VirS and its cognate DNA operator region in a controlled biological context devoid of other bacterial proteins. We observed a strong concentration-dependent reduction of SEAP production upon alpibectir treatment ($IC_{50} = 0.001$ mg/L) (Fig. 3A). Conversely, we showed that synthetic gene circuits designed to sense interactions between EthR2 or EthR and their respective cognate DNA operator were modestly impacted by alpibectir ($IC_{50} = 0.27$ mg/L) or not affected at all, respectively (SI Fig. 5). These findings are consistent with the low change in transcriptomic level for *ethA2* and *ethA*.

The direct binding between alpibectir and VirS was confirmed using differential scanning fluorimetry, a method that assesses the impact of a ligand on the temperature at which a protein unfolds[26]. The assay revealed a concentration-dependent thermostabilizing effect of alpibectir on VirS (Fig. 3B, C). Notably, the most pronounced stabilization, resulting in a 10 °C increase in the melting temperature of VirS, was achieved in the presence of an equimolar concentration of alpibectir (Fig. 3C).

To improve our understanding of the direct interaction between VirS and alpibectir, we co-crystallized their complex and refined the crystal structure at a resolution of 1.49 Å (PDB ID 8RCX; SI Table 2). Alpibectir resides within a fully enclosed and predominantly hydrophobic cavity of VirS. The carbonyl group of alpibectir establishes 2 hydrogen bonds with the Nδ2 side chain amides of Asn11 and Asn149 (Fig. 3D and Fig. 3E). Its piperidine moiety is anchored on one side through the Ile80 side chain and on the opposite side by the Tyr129 and Ser146 side chains. The isoxazoline ring engages prominently with the Met153 side chain, while Gly142 provides the necessary spatial accommodation for the trifluoromethyl group adjacent to the $CH_2$-$CH_2$ aliphatic chain of the compound. This trifluoromethyl group is further stabilized by the side chains of Ile7, Leu49, Phe52, Val53, and Ser146. The second trifluoromethyl group is anchored by the side chains of Ile101, Leu114, Met153, and Phe197.

Conclusively, the integration of the transcriptomic and the structural data presented herein indicates that the upregulation of the *mymA* operon is caused by the direct physical interaction of alpibectir with the putative ligand-binding domain of VirS.

To assess whether the specificity of alpibectir for VirS also translated *in bacterio*, we tested its impact on a *virS* mutated strain of *M. tuberculosis* (strain 8 F with mutation Cys278Arg in VirS). Consistent with published data, the loss of function mutation in *virS* resulted in a slight decrease in susceptibility to Eto[21,27] (Fig. 4). Alpibectir did not restore Eto activity in strain 8 F, confirming the specificity of its activity through VirS, and highlighting the central role of this transcriptional regulator in controlling the boost of the bioactivation of Eto mediated by alpibectir.

Strikingly, the mutation in *virS* (strain 8 F) not only abrogated the Eto-boosting activity of alpibectir, but it also suppressed the intrinsic antibacterial activity of alpibectir, leading to the conclusion that both the boosting and the intrinsic activities of alpibectir are under the control of VirS.

## Alpibectir has distinct intrinsic and ethionamide boosting actions

To delineate the intrinsic and the Eto-boosting activities of alpibectir, we investigated additional mutants in the *virS-mymA* regulon. Insertion of a transposon into the *mymA* open reading frame (Rv3083::Tn), which abrogates the expression of the entire operon without affecting the expression of VirS, resulted in the loss of both the Eto-boosting and intrinsic antibacterial activity of alpibectir (SI Table 3 and SI Fig. 7). These findings rule out the possibility that the intrinsic antibacterial activity of alpibectir results from the expression of any VirS-regulated gene outside the *virS-mymA* regulon. They align with the RNAseq experiments, which demonstrate that alpibectir predominantly modulates the expression level of the *virS-mymA* regulon. This suggests that both the Eto-boosting effect and the intrinsic antibacterial activity of alpibectir arise from upregulation of the *mymA* operon. Remarkably, the inactivation of *adhD*, the fourth gene in the operon, by homologous recombination (which, in addition, might impact the alpibectir-mediated overexpression of the genes downstream of *adhD*), resulted in a complete loss of the intrinsic antibacterial activity of alpibectir, without affecting its Eto-boosting effect (Fig. 5). Future studies aimed at generating clean, non-polar deletions of genes within the *mymA* operon and performing gene complementation experiments will be necessary to fully elucidate the genetic basis of the intrinsic activity of alpibectir.

In conclusion, by stimulating the expression of the *mymA* operon, alpibectir leads to two distinct outcomes: on one hand, the overproduction of the MymA protein potentiates the bioactivation of Eto; on the other hand, the overexpression of *adhD* and/or of its downstream genes directly or indirectly affects the growth of *M. tuberculosis* in vitro.

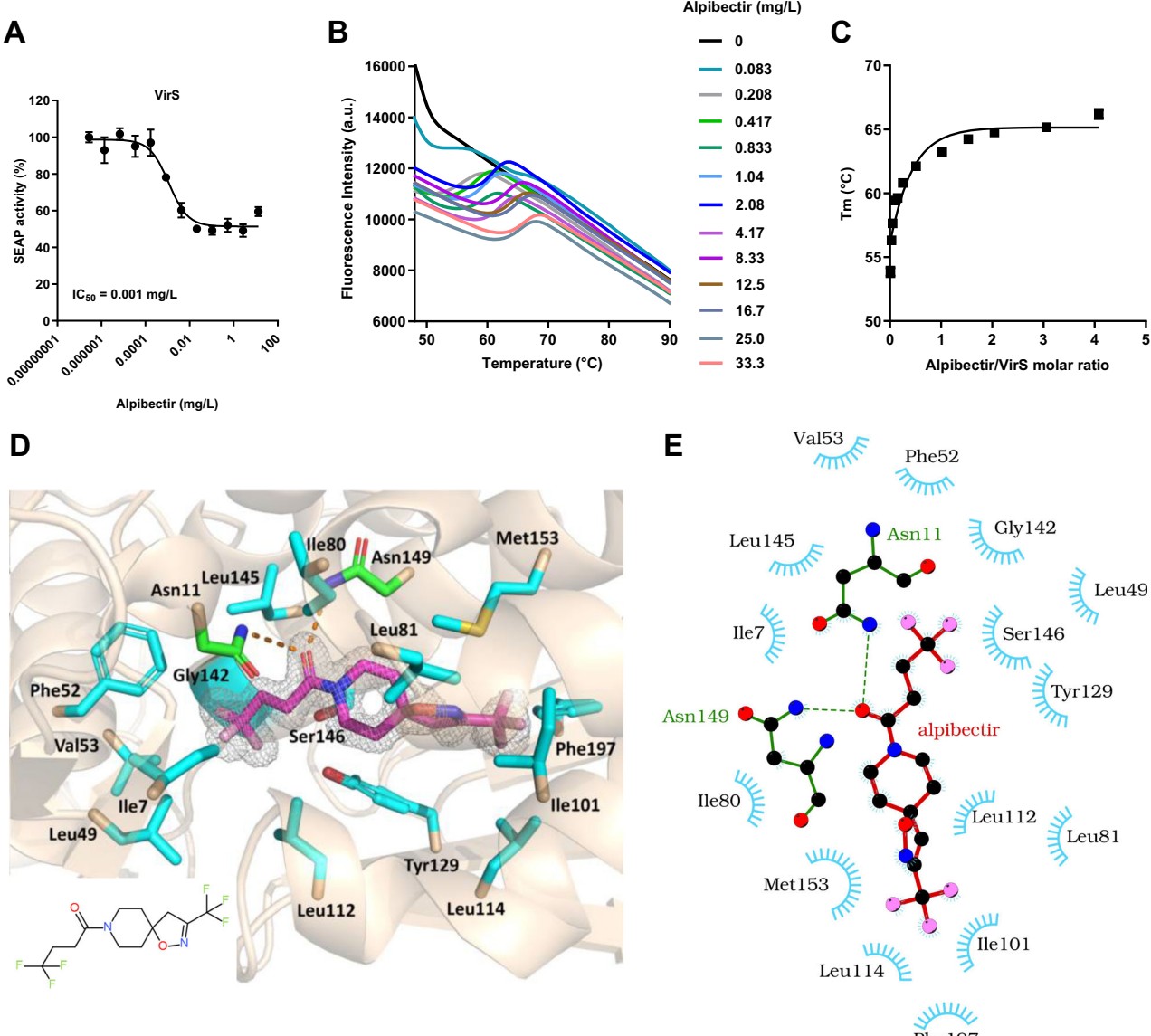

**Fig. 3 | Alpibectir inhibits VirS by binding to it. A** Concentration-dependent inhibition of the binding of VirS to its cognate DNA operator region in the presence of varying concentrations of alpibectir. DNA binding was measured by the expression of the SEAP reporter gene. The $IC_{50}$ value is indicated. Data points represent the mean ± standard deviation of 2 technical replicates. Source data are provided as a Source Data file. **B** The thermal denaturation of VirS was monitored using SYPRO Orange in the presence of increasing concentrations of alpibectir. The fluorescence signal was plotted as a function of temperature. α-latalbumin was used as a negative control (SI Fig. 6). Source data are provided as a Source Data file. **C** The melting point (Tm) of VirS was obtained using the first-derivative of the fluorescence as a function of temperature (-dF/dT) and was plotted as a function of ligand/protein molar ratio. Data points represent the mean ± standard deviation of 5 technical replicates. **D** A close-up view of the VirS-ligand binding domain showing the interactions between alpibectir and the protein. The VirS protein is shown as a transparent cartoon. Alpibectir and interacting VirS amino acid side chains are

represented by sticks with nitrogen, oxygen, fluorine, and sulfur atoms in blue, red, pink, and yellow, respectively, and carbon atoms in magenta (alpibectir) or cyan (protein). The two asparagine amino acids, Asn11 and Asn149, which establish hydrogen bonds with the drug are distinguished by their green-colored carbon atoms. Gly142 is located by coloring (cyan) its Cα atom onto the protein ribbon. The interacting residues are labelled, and the hydrogen bonds are represented by dashed orange lines. Electron density of $F_o$-$F_c$ omit map contoured at 2.0σ is shown with a grey mesh around alpibectir. The image was produced by the PyMOL Molecular Graphics System (version 2.4.0, Schrödinger, LLC). The inset shows the structural formula of alpibectir. **E** A 2D schematic diagram of alpibectir-VirS interactions generated by the program LigPlot+[62]. Oxygen, nitrogen, carbon, and fluorine atoms are in red, blue, black, and pink, respectively. Hydrogen bonds are shown as green dotted lines with the distance indicated, while the blue spoked arcs represent protein residues in contact with the alpibectir molecule.

## AlpE is bactericidal against *M. tuberculosis*

Next, we assessed the speed of the anti-TB activity of AlpE using time-kill curves. At a concentration near its MIC (2.5 mg/L), Eto alone exhibited a constant yet relatively slow bactericidal activity in vitro, resulting in an approximate 4-$\log_{10}$ reduction of CFU in 7 days, never reaching the limit of detection of $10^2$ CFU/mL. After this 7-day period, rapid regrowth was observed in the culture at day 21 (Fig. 6A). At a tenfold higher Eto concentration (25 mg/L), a similar initial rate of

CFU reduction was observed, reaching the detection limit after 14 days. Nonetheless, the onset of regrowth was observed by 21 days (Fig. 6A).

Conversely, combining Eto 2.5 mg/L with alpibectir resulted in rapid bactericidal activity, driving bacterial counts down to the detection limit ($10^2$ CFU/mL) within 7 days, with no regrowth observed up to 21 days. In comparison, as well-documented, INH at 10 times its MIC (2.5 mg/L) failed to prevent bacterial regrowth.

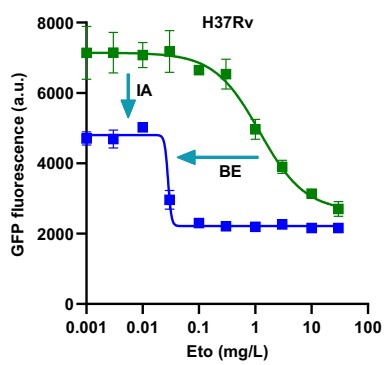
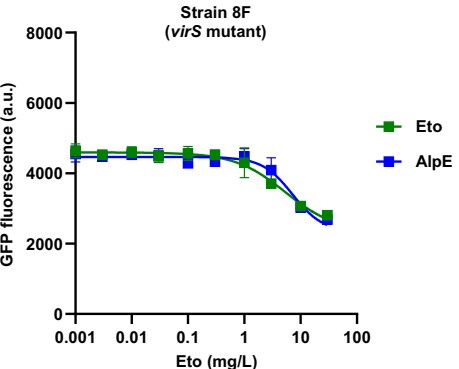

**Fig. 4 | Mutation Cys278Arg in VirS (strain 8 F) abrogates the Eto boosting activity and the intrinsic activity of alpibectir.** Eto was titrated, alpibectir was kept at a fixed concentration of 0.33 mg/L. AlpE alpibectir/Eto combination, IA

Alpibectir intrinsic activity, BE Ethionamide boosting activity. Results shown are representative of 3 biological replicates. Data points represent the mean ± standard deviation of 2 technical replicates. Source data are provided as a Source Data file.

In comparison to the intrinsic growth-inhibitory activity shown in Fig. 5, no bactericidal activity was observed when alpibectir was used alone at concentrations up to 0.3 mg/L, suggesting that it affects bacterial fitness rather than viability. Similarly, AlpE was equally effective in killing the H37Rv strain mutated in *adhD*, indicating that the intrinsic effect of alpibectir does not contribute to the killing effect of the combination nor does it accelerate it (Fig. 6B).

## Alpibectir reduces the frequency of spontaneous resistance to Eto

We then evaluated whether the enhanced killing effect of AlpE impacts the frequency of resistance (FoR) of *M. tuberculosis* H37Rv compared to Eto alone. INH and moxifloxacin (Mox) were used as references. The FoR to INH and Mox (at 10-fold and 4-fold agar MIC, respectively) were consistent with literature data[28,29], with values around $10^{-6}$ and $10^{-8}$, respectively. Eto at 2.5 mg/L (agar MIC) resulted in a FoR of $5 \times 10^{-7}$. Remarkably, the combination of 0.012 mg/L alpibectir with 2.5 mg/L Eto prevented the emergence of resistant colonies, indicating a FoR below the detection limit of $10^{-9}$ (Table 1).

In the presence of alpibectir, resistant mutants were only obtained at lower Eto concentrations, ranging from 0.375 to 1 mg/L (SI Table 4). Detailed analysis of these low-resistance clones revealed mutations in genes previously associated with Eto resistance, namely *ethA*, *mymA*, or *virS*. Four clones had simultaneous mutations in both *ethA* and *mymA* (SI Table 5). Subsequently, the susceptibility of 14 resistant clones to various anti-TB drugs was tested. All clones were sensitive to moxifloxacin, linezolid, ethambutol, bedaquiline, rifampicin, and pretomanid, indicating the absence of a global or specific mechanism of cross-resistance between AlpE and other TB drugs (SI Fig. 8).

## Alpibectir restores Eto and Pto activity in *ethA* mutants

Pto is the propyl analogue of Eto (SI Fig. 9). Both antibiotics are considered interchangeable for MDR-TB treatment regimens and for treatment of tuberculous meningitis in adults and children[5,30].

Since the efficacy of the prodrugs depends on their bioactivation level, it is crucial to assess whether the overexpression of *mymA* mediated by alpibectir potentiates Pto and Eto to the same extent. The antibacterial activities of AlpE (alpibectir/ethionamide) and alpibectir/prothionamide were compared against the H37Rv strain and various isogenic clones with mutations in the *ethR-ethA* regulon, the *virS-mymA* regulon, or both simultaneously (SI Fig. 10). The concentration-response curves for Eto alone ($IC_{50} = 1.18$ mg/L) and Pto alone ($IC_{50} = 0.31$ mg/L) were similar against H37Rv, with Pto being slightly more potent. The inclusion of alpibectir similarly enhanced the efficacy of Eto and Pto, shifting the $IC_{50}$ values to 0.03 mg/L for both (SI Table 3). Consistent with published data, the potency of Eto and Pto

was compromised by the inactivation of *ethA* (Rv3854::Tn) or *mymA* (Rv3083::Tn)[27]. Complete resistance to both thioamides was observed in the double mutant strain CD6R1 (*ethA⁻/mymA⁻*), which lacks both bioactivation pathways[27]. Remarkably, alpibectir restored the sensitivity of both Eto and Pto in the *ethA*-deficient strain at levels equivalent to those observed in H37Rv. This suggests that the enhanced expression of the *mymA* operon induced by alpibectir achieves maximum activation of both prodrugs, irrespective of the *ethA* expression level. As expected, alpibectir did not enhance Pto potency in the *mymA*-deficient strain (Rv3083::Tn). This set of results collectively demonstrates that alpibectir uniformly boosts the activity of both thioamides through stimulation of the *virS-mymA* regulon, even in strains that lack EthA production.

## Alpibectir boosts the potency of Eto and Pto sulfoxide metabolites

Upon oral administration, human flavin-containing monooxygenases (FMO) convert some of the circulating Eto or Pto into sulfoxide derivatives Eto-SO[30–33] and Pto-SO (manuscript in preparation), respectively. Interestingly, these intermediates are as active as their corresponding parental prodrug. As already suggested by Vannelli and colleagues[34] and confirmed by us[18,34], both Eto-SO and Pto-SO still require *M. tuberculosis* FMOs such as EthA to be further transformed and become effective. Interestingly, addition of alpibectir to Eto-SO and Pto-SO boosted their respective potency against H37Rv in a similar amplitude to the one observed with the respective parental compounds (SI Table 3), showing that MymA, like EthA, is able to finalize the transformation of the sulfoxide derivatives into active compounds. Finally, the reduced potency of Eto-SO and Pto-SO observed against an EthA-deficient mutant was fully restored by the addition of alpibectir. This observation implies that both the parental prodrugs and their respective SO derivatives, whether they are produced by the liver or the lung FMOs, will be boosted by alpibectir.

## Alpibectir boosts Eto in recombinant strains over-expressing *inhA*

Although the intracellular activation of Eto and INH occurs through distinct and independent enzymatic pathways, it results in the formation of analogous covalent adducts between each drug and nicotinamide adenine dinucleotide (NAD). Both adducts subsequently inhibit the enoyl reductase InhA, a key enzyme in mycolic acid biosynthesis[31–33]. Here, we assessed the antibacterial activity of Eto alone or in combination with alpibectir on *M. tuberculosis* overexpressing *inhA* (SI Fig. 11). When *inhA* is expressed at basal level, alpibectir decreased the $IC_{50}$ of Eto from 0.07 to 0.01 mg/L, consistent with the observed effect on the H37Rv strain (Fig. 1). Overexpression of

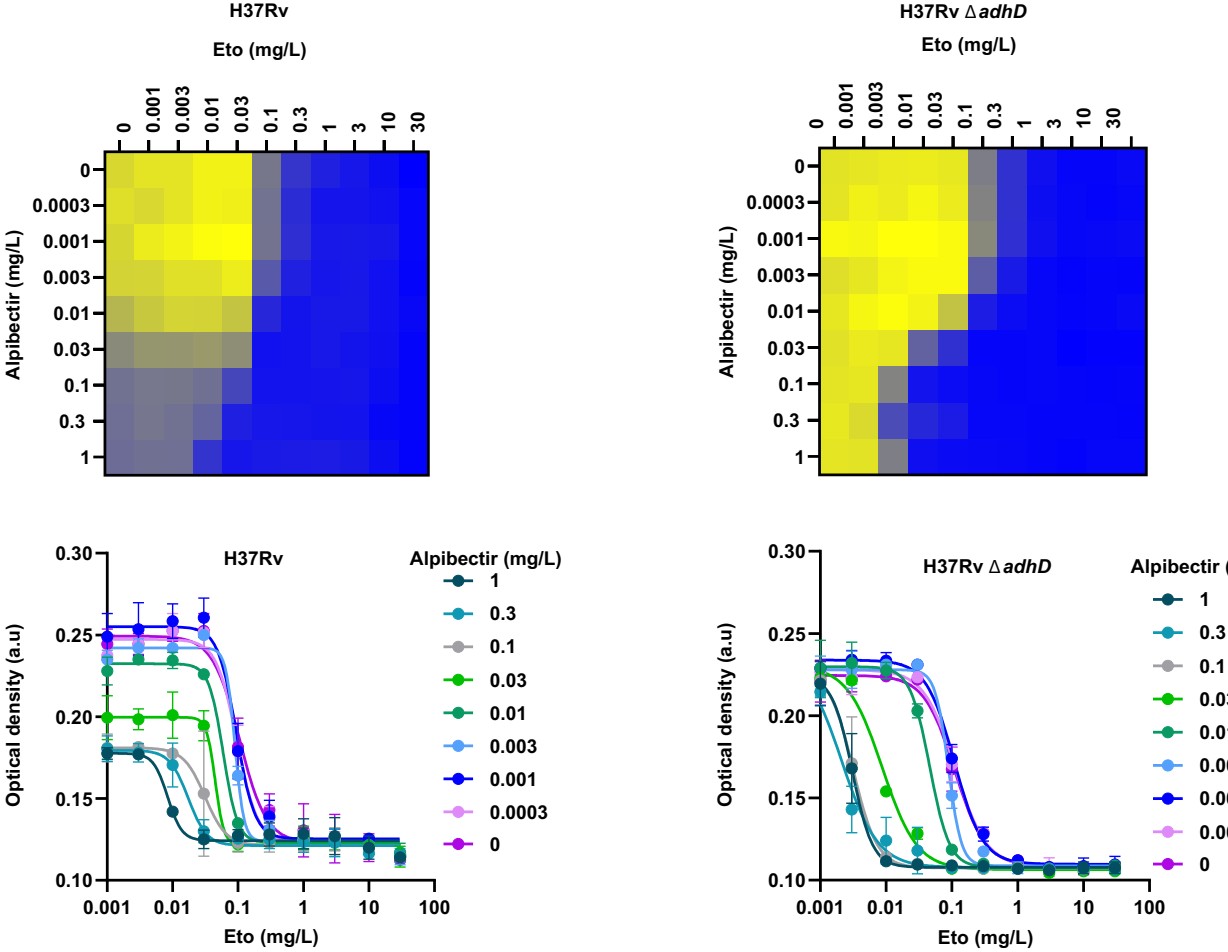

**Fig. 5 | Effect of *adhD* on the susceptibility of *M. tuberculosis* to the alpibectir/ Eto (AlpE) combination using optical density.** The heatmaps (blue to yellow) show the change in optical density. Data points represent the mean ± standard deviation of 2 technical replicates. Results shown are representative of 3 biological replicates. Source data are provided as a Source Data file.

*inhA* increased the $IC_{50}$ of Eto from 0.07 to 1.4 mg/L. Notably, treatment of the *inhA*-overexpressing strain with AlpE decreased the $IC_{50}$ of Eto from 1.4 to 0.3 mg/L. This shows that alpibectir is potent enough to enhance the efficacy of Eto, even when its target, InhA, is significantly overproduced. This also suggests that the bioactivation of ethionamide by MymA results in the formation of an NAD-adduct, similar to the process observed with EthA. Addition of alpibectir did not impact the MIC of INH, confirming that although InhA is targeted by both INH and Eto, alpibectir is specific for the activation of Eto (SI Fig. 12).

## AlpE inhibits growth of MDR-TB, including *ethA* and *inhA* mutants

The in vitro susceptibility of *M. tuberculosis* to Eto alone or the combination (AlpE) was assessed using liquid cultures (BACTEC™ MGIT™ 960). The activity is expressed as the MIC to suppress the growth of the strains. First, the activity was assessed on the standard laboratory strain H37Rv. The MIC was 1 mg/L for Eto alone. In the presence of alpibectir (tested at 0.017, 0.05, or 0.1 mg/L), the MIC of Eto decreased significantly to ≤ 0.031 mg/L, once more demonstrating the strong Eto potentiation by the booster (SI Table 6). The presence of alpibectir alone did not inhibit the growth of the bacteria at any of the 3 concentrations tested. Next, we looked at a panel of INH-resistant *M. tuberculosis* clinical strains (MIC > 0.1 mg/L) that also exhibited cross-resistance to Eto (MIC > 5 mg/L). Among the 21 strains selected, 12 had mutations in the *inhA* promoter and 16 had a mutation in *katG* (S315T). Most strains (18/21) were also resistant to rifampicin (Table 2).The

ability of alpibectir to potentiate Eto activity was evaluated on these strains using alpibectir at 0.017 mg/L or 0.05 mg/L, combined with Eto at 0.8 mg/L. This concentration corresponds approximately to 1/6th of the clinical breakpoint (5 mg/L) above which strains are classified as resistant to Eto[35]. This sub-inhibitory Eto concentration was chosen to allow detection of changes in drug susceptibility of the strains to Eto in the presence of alpibectir. The two alpibectir concentrations enable the detection of a dose-dependent effect for certain strains with the higher concentration capturing the full extent of Eto potentiation.

Remarkably, 11 (92%) of the 12 $INH^R/Eto^R$ strains mutated in the *inhA* promoter were re-sensitized to Eto 0.8 mg/L in the presence of 0.05 mg/L alpibectir and 9 strains (75%) were re-sensitized with 0.017 mg/L alpibectir. All 9 $INH^R/Eto^R$ clinical isolates without mutation in the promoter of *inhA* were sensitive to 0.8 mg/L Eto in the presence of 0.017 mg/L alpibectir. Alpibectir alone did not inhibit the growth of any of the strains up to the highest tested concentration of 0.05 mg/L. Overall, 95% of INH/Eto cross-resistant isolates demonstrated sensitivity to the combination of Eto (0.8 mg/L) and alpibectir (0.05 mg/L). Only one isolate (13M2489, lineage 2) was unresponsive to the alpibectir/Eto combination. Whole genome sequencing of this strain revealed a large deletion in the *virS/mymA* regulon, consistent with the proposed mode of action of alpibectir. This mutation is extremely rare among lineage 2 strains. In the CRyPTIC database, which includes data from 12,287 *M. tuberculosis* isolates, 4295 strains belong to lineage 2, and none of these exhibits a complete deletion of the *virS-mymA* regulon. It is worth noting, however, that *M. tuberculosis* sublineage 4.8

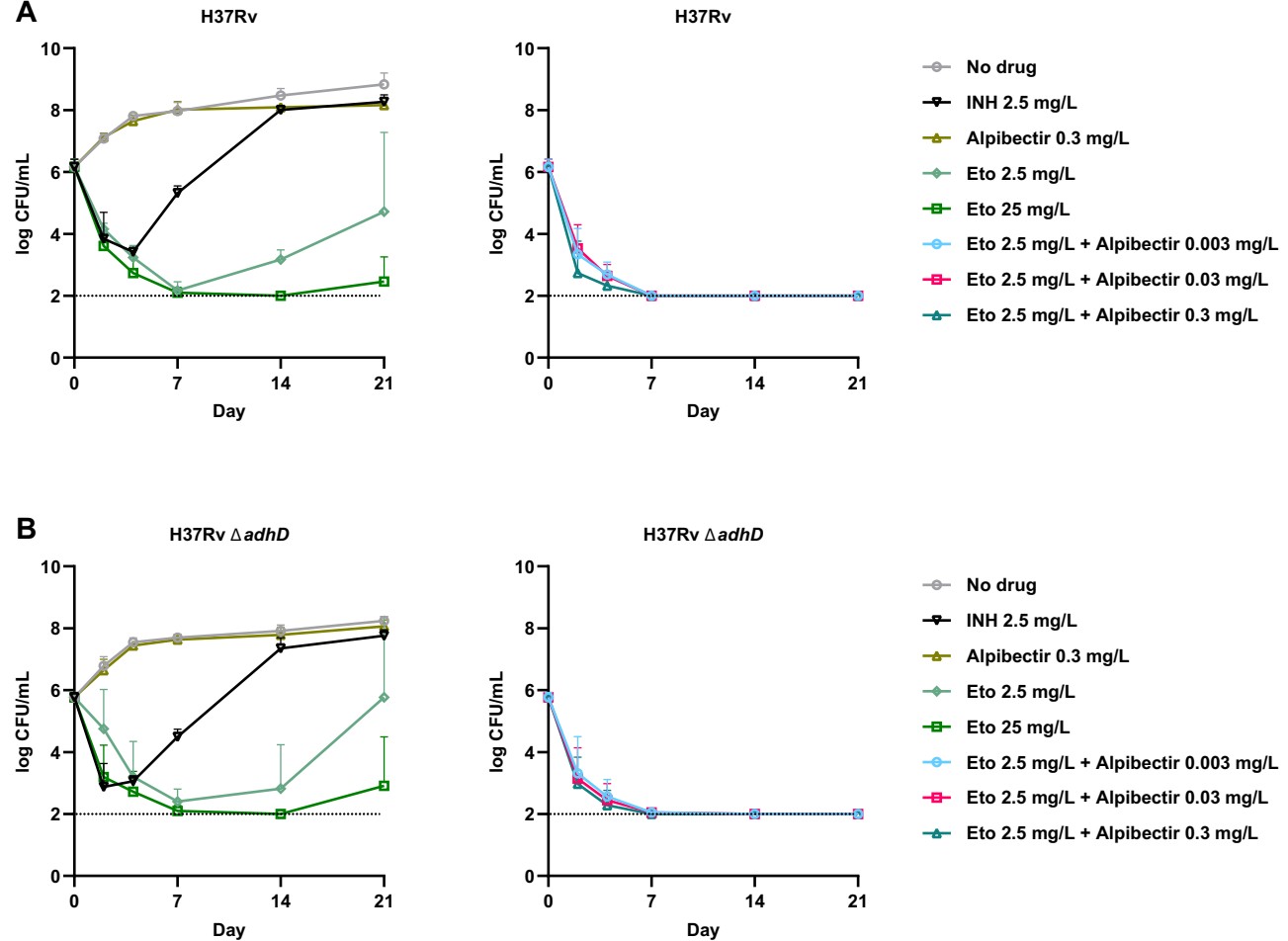

**Fig. 6 | Alpibectir accelerates the bactericidal kinetics of Eto.** Survival kinetics of *M. tuberculosis* H37Rv (**A**) and H37Rv ΔadhD (**B**) treated with Eto alone or combined with alpibectir, or with INH alone. *M. tuberculosis* was cultured in the absence (*No drug* control) or presence of drugs at the indicated concentrations. At each time point, an aliquot of the cultures was plated on antibiotic-free media for CFU counting. The dotted line indicates the limit of detection of CFU counts. Data points represent the mean ± standard deviation of 3 biological replicates. Source data are provided as a Source Data file.

**Table 1 | Frequency of spontaneous resistance of *M. tuberculosis* H37Rv to INH, Mox, or Eto alone or combined with alpibectir (AlpE)**

| Frequency of resistance to | | | |
|---|---|---|---|
| INH (2 mg/L) | Mox (0.32 mg/L) | Eto (2.5 mg/L) | AlpE (Eto (2.5 mg/L) + alpibectir (0.012 mg/L)) |
| $3 \times 10^{-6}$ | $4 \times 10^{-8}$ | $5 \times 10^{-7}$ | $<1 \times 10^{-9}$ |

Source data are provided as a Source Data file.

naturally harbors a similar large deletion encompassing the *virS-mymA* regulon[36], rendering these strains intrinsically unresponsive to alpibectir. Accordingly, particular caution will be necessary in regions where this sublineage is prevalent, to avoid lowering the Eto dose to sub-MIC levels in patients infected with such strains.

**AlpE is active in vivo in a 4-week acute murine model of TB**
The ability of alpibectir to potentiate the activity of Eto was evaluated in an acute mouse model of TB[37,38]. Mice were intravenously (IV) infected with *M. tuberculosis* H37Rv and then treated for 28 days with various doses of Eto, alone or in combination with alpibectir. In untreated controls, lung bacillary load increased by +3.5 $\log_{10}$ CFU ($p < 0.0001$) relative to the initial load of 5.4 $\log_{10}$ CFU at pre-

treatment. Oral administration of INH (25 mg/kg q24h, reflecting the human equivalent dose[39]) reduced the bacillary burden to 4.4 $\log_{10}$ at D28 ($p = 0.06$), equivalent to a 1 $\log_{10}$ reduction from the initial load (Fig. 7, SI Table 7A).

Evaluation of Eto alone (5–200 mg/kg) showed a dose-dependent improvement in treatment efficacy; however, only the 200 mg/kg dose led to a statistically significant 1 $\log_{10}$ CFU reduction compared with pre-treatment (SI Table 7B).

In mice treated solely with alpibectir at 0.1, 0.5, or 1.6 mg/kg, lung CFU counts at D28 did not fall below the initial load. Nevertheless, at 1.6 mg/kg, alpibectir produced lower CFU counts than at 0.5 or 0.1 mg/kg, and was 2.1 $\log_{10}$ below the untreated control, consistent with the intrinsic growth-inhibitory effect of alpibectir observed in vitro.

Combinations of alpibectir and Eto resulted in greater bacterial load reduction: whereas Eto at 5 mg/kg alone was only modestly active (7.8 $\log_{10}$ CFU), adding alpibectir at 0.1, 0.5, or 1.6 mg/kg reduced the CFU counts by an additional 1.5 ($p = 0.0043$), 2.8 ($p < 0.0001$), and 2.3 ($p < 0.0001$) $\log_{10}$, respectively (Fig. 7). Moreover, Eto doses of 15 or 50 mg/kg combined with 0.5 or 1.6 mg/kg alpibectir decreased the CFU counts by more than 1 $\log_{10}$ relative to pre-treatment levels. In combination with alpibectir, Eto at 15 mg/kg achieved maximal bactericidal activity (Emax). Increasing the ethionamide dose to 50 mg/kg provided no additional benefit, suggesting that the pharmacological target, InhA, is already saturated at 15 mg/kg. In contrast, the lowest

**Table 2 | Impact of alpibectir on the susceptibility to Eto on a panel of Eto-resistant and INH-resistant clinical strains**

| M. tuberculosis strains | | INH 0.1 mg/L | Rif 1 mg/L | Eto MIC mg/L | Alpibectir 0.05 mg/L | Eto 0.8 mg/L + alpibectir 0.017 mg/L | Eto 0.8 mg/L + alpibectir 0.05 mg/L | inhA-promoter | InhA (Rv1484) | KatG (Rv1908) | EthA (Rv3854c) | VirS (Rv3082c) | MymA (Rv3083) |
|---|---|---|---|---|---|---|---|---|---|---|---|---|---|
| **Strains Eto-R (MIC ≥ 5) and INH-R (MIC ≥ 0.1 mg/L) mutated in inhA-Promoter or ORF** | | | | | | | | | | | | | |
| 08MYO593 | Beijing | G | G | ≥20 | G | NG | NG | C −15 T | wt | S315T/R463L | wt | wt | wt |
| 08MYO891 | LAM9 | G | G | ≥20 | G | G | NG | C −15 T | wt | S315T | Y84D | wt | wt |
| 11MYO210 | H3 | G | G | 10 | G | NG | NG | C −15 T | wt | wt | wt | wt | wt |
| 13MYO376 | T1 | G | NG | 5 | G | NG | NG | C −15 T | wt | wt | wt | wt | wt |
| 07MYO788 | Beijing | G | G | ≥20 | G | NG | NG | C −15 T | wt | S315 T/R463L | wt | wt | wt |
| 13MY2475 | Afri1 | G | G | 5 | G | NG | NG | C-15T | wt | wt | I337V | wt | wt |
| 08MY1602 | LAM9 | G | G | ≥256 | G | NG | NG | C-15T | wt | S315T | R239G | wt | wt |
| 10MYO981 | Beijing | G | G | ≥20 | G | NG | NG | C-15T | wt | S315T/R463L | wt | wt | wt |
| 08MYO057 | CAS1 | G | G | 64 | G | G | NG | T-8C | wt | S315T/R463L | A247fs | wt | wt |
| 09MYO391 | Beijing | G | G | 20 | G | NG | NG | T-8C | wt | S315T/R463L | T314I | wt | wt |
| 07MYO066 | Beijing | G | G | 5 | G | NG | NG | T-8C | wt | S315 T/R463L | R463fs | wt | wt |
| 09MYO467 | Beijing | G | G | 5 | G | NG | NG | wt | I194T | W149C/R463L | wt | wt | wt |
| 13MY2489 | Beijing | G | NG | ≥20 | G | G | G | C-15T | wt | R463L | wt | Del | part-Del |
| **Strains Eto-R (MIC ≥ 5) and INH-R (MIC ≥ 0.1 mg/L) with WT inhA-Promoter and ORF** | | | | | | | | | | | | | |
| 07MY1004 | Beijing | G | G | 32 | G | NG | NG | wt | wt | S315T/R463L | T314I | wt | wt |
| 08MY1099 | Beijing | G | G | 5 | G | NG | NG | wt | wt | S315T | wt | ND | ND |
| 07MY1001 | Beijing | G | G | 32 | G | NG | NG | wt | wt | S315T/R463L | T314I | wt | wt |
| 07MY1166 | Beijing | G | G | 64 | G | NG | NG | wt | wt | S315T/R463L | P378L | wt | wt |
| 07MY1281 | Beijing | G | G | 16 | G | NG | NG | wt | wt | S315T/R463L | W289* | wt | wt |
| 09MY1304 | Beijing | G | G | 32 | G | NG | NG | wt | wt | S315T/R463L | Y147* | wt | wt |
| 10MYO992 | Beijing | G | G | ≥10 | G | NG | NG | wt | wt | S315T/R463L | K37fs | wt | wt |
| 08MYO559 | LAM9 | G | G | 5 | G | NG | NG | wt | wt | S315T | Pro209fs | G96S | wt |

Rif rifampicin, wt wild-type, ND not determined, G (red) growth observed, NG (green) no growth observed.

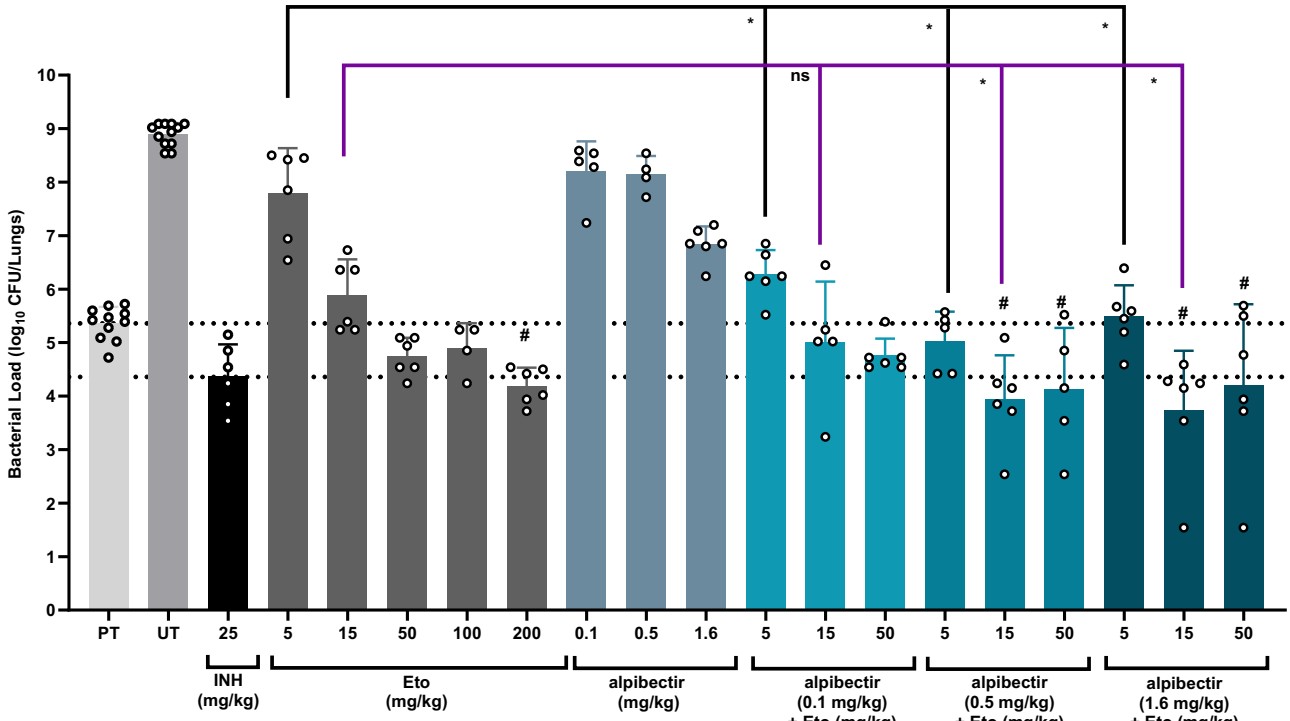

**Fig. 7 | Lung CFU counts assessed at D28 in infected BALB/c mice treated with Eto and alpibectir.** # Lung CFU counts that are statistically significantly lower than pre-treatment ($p < 0.05$; exact values are given in SI Table 7A); * Lung CFU counts that are statistically different compared to the groups without alpibectir ($p < 0.0001$ except for Eto 5 mg/kg *vs*. Alpibectir 0.1 mg/kg + Eto 5 mg/kg whereby $p = 0.004$); ns non-significant difference; PT pre-treatment group; UT untreated control group. The limit of detection was at 1.24 log$_{10}$ CFU. The pre-treatment and untreated groups contain data from 11 and 12 mice, respectively. The treated groups contain data from 4, 5, or 6 mice. Each treatment group is colored differently. Data points represent the bacterial load per mouse. The means ± standard deviations are given. Statistical analyses were performed using one-way ANOVA (two-sided) with Dunnett's test for multiple comparisons. Source data are provided as a Source Data file.

tested dose of Eto (5 mg/kg) failed to reach this level of activity, indicating that the minimal effective Eto dose in the presence of alpibectir lies between 5 and 15 mg/kg in this murine model.

A sigmoidal dose-effect analysis (SI Fig. 13) indicated that an Eto dose of ±90 mg/kg was required to achieve a 1 log$_{10}$ CFU reduction compared with the initial bacillary load, matching the effect of 25 mg/kg INH. In contrast, in the presence of alpibectir at 0.1, 0.5, or 1.6 mg/kg, the Eto dose needed for this same reduction fell to 33.0, 7.5, and 8.5 mg/kg, respectively. Notably, alpibectir at doses ≥ 0.5 mg/kg lowered the required Eto dose by tenfold (SI Fig. 13). Strikingly, alpibectir alone at 1.6 mg/kg prevented mortality ($p = 0.0001$), consistent with its impact on lung CFU counts (SI Fig. 14B) and with its intrinsic growth-inhibitory activity observed in vitro. Although alpibectir at 0.1 and 0.5 mg/kg also had a significant yet moderate impact on survival (17%), combining alpibectir with as little as 5 mg/kg Eto completely prevented mortality (SI Fig. 14C).

## Discussion

TB remains a significant global health problem[1]. In 2023, an estimated 10.8 million people fell ill with TB and 1.25 million died. Increasing resistance of *M. tuberculosis* to currently available anti-TB drugs threatens the progress made toward the global goal of TB elimination in recent years. Designed to take control of specific gene expression in the *M. tuberculosis* pathogen, alpibectir is a first-in-class compound tested in human[22].

Alpibectir is the culmination of more than two decades of research aimed at stimulating the expression of genes involved in the bioactivation of the prodrugs Eto and Pto[15]. The first generation of compounds emerged from a target-based approach designed to inhibit EthR, the transcriptional repressor of *ethA*, which encodes the principal enzyme activating Eto in *M. tuberculosis*[10]. The lead molecule, BDM41906, over-sensitized bacteria to Eto by 20-fold in vitro but, as anticipated, was ineffective against clinical strains harboring *ethA* mutations, commonly observed in clinical practice. A second series of compounds emerged serendipitously from a structural diversification campaign around BDM41906. We observed that two families of compounds lost all affinity for EthR but unexpectedly still boosted Eto activity. SMARt420 and SMARt751, the lead representatives of these families, were found to stimulate two alternative pathways of Eto-activation: EthR2-EthA2[20] and VirS-MymA[21], respectively. The key finding of these studies was that these compounds not only enhanced the potency of Eto against pan-sensitive *M. tuberculosis* strains but also restored full sensitivity to strains carrying *ethA* mutations, the predominant allele conferring Eto resistance in clinical settings.

Alpibectir has emerged from an intense optimization campaign aimed at targeting VirS to enhance the expression of the *mymA* operon. Through a series of biochemical and biophysical experiments, we collected evidence that alpibectir directly interacts with VirS, modulating its DNA-binding capabilities. The co-crystal structure of the alpibectir-VirS complex revealed how alpibectir binds to the VirS ligand-binding domain rather than the helix-turn-helix (HTH) region, indicating an allosteric mechanism of action rather than steric interference at the regulator-DNA interface. The *in-cell* transcription regulator-DNA binding assay further confirmed that alpibectir modulates VirS-driven activity, which was also shown to stimulate the expression of the *mymA* operon in our transcriptomics experiments[40].

Unexpectedly, alpibectir alone exhibited a concentration-dependent inhibitory effect on bacterial growth. Transcriptional analysis showed that alpibectir specifically stimulates the *mymA* operon. Its selectivity in controlling *mymA* expression via interaction with VirS

suggests that the compound's intrinsic activity depends on one or more genes within this operon. Supporting this hypothesis, the antibacterial effect of alpibectir was completely abolished in *virS* or *mymA* mutants. Further, we demonstrated that the intrinsic antibacterial effect of alpibectir can be separated from its ability to enhance Eto activity by genetically disrupting *adhD*, the fourth open reading frame in the seven-member *mymA* operon, thereby altering alpibectir-mediated overexpression of the remaining three downstream genes.

Interestingly, alpibectir's intrinsic activity, demonstrated in vitro, also manifested in vivo, as evidenced by reduced bacterial growth and prevention of mortality in an acute mouse TB model. While this intrinsic effect is largely masked by Eto co-treatment in vitro, its contribution to limiting bacterial growth in vivo is evident. Additional in vivo experiments involving the *adhD* mutant will be necessary to further elucidate these mechanisms.

Due to the *ethR/ethA*-independent bioactivation mechanism, alpibectir overcomes Eto resistance and keeps potent activity against drug-resistant strains. Alpibectir restores the activity of Eto in *ethA* mutants, which again confirms that its main mode of action is to bioactivate Eto through the action of the EthA homologous enzyme MymA. Eto bioactivation by EthA and MymA remains mechanistically unresolved, and their relative catalytic contributions are unknown. Nonetheless, alpibectir-driven upregulation of *mymA* increases Eto potency relative to baseline, consistent with increased formation of the activated species. Moreover, the growth protection conferred by *inhA* overexpression, even in the presence of alpibectir, supports the view that the MymA-catalyzed product is the Eto-NAD adduct targeting InhA, aligning with the established target-engagement model. Interaction with a relatively low amount of regulator protein (VirS) leading to the production of a high amount of effector enzyme (MymA) and in turn leading to the expected phenotype may be one of the greatest benefits of using drugs targeting bacterial regulatory systems, like alpibectir.

At the outset of this millennium, INH was often acknowledged as the cornerstone of the first-line regimen against TB due to its high early bactericidal activity (EBA), achieving a remarkable 80 to 90% reduction in viable bacilli in just 2 days[41,42]. However, INH mono-resistance is the most prevalent form of drug-resistant TB worldwide (besides streptomycin resistance), with estimates rising to 7% among new TB cases and 8 to 11% among previously treated TB cases[43,44], a total of approximately 1 million cases annually. INH mono-resistant TB is associated with a higher risk of acquiring further drug resistance, evolving towards MDR-TB, and dramatically reducing treatment success rate[45–47]. Consequently, agents with an EBA comparable to that of INH, sharing the same mode of action without being perturbed by existing INH resistance mechanisms, would certainly be a key asset to be associated with new therapeutic regimens. Almost all Eto-resistant clinical strains with *inhA* promoter mutations were re-sensitized to Eto in the presence of alpibectir, demonstrating that this compound is able to potentiate the activity of Eto to such an extent that it compensates for InhA overexpression. In addition, due to the use of a *katG*-independent activation mechanism, AlpE is active on *katG*-mutated INH-resistant strains. Combination with alpibectir lowered the concentration of Eto required for cidality and increased the rate of killing, AlpE being as rapidly bactericidal as INH and much more efficient in controlling regrowth in vitro. Finally, the presence of alpibectir drastically reduced the frequency of resistance of Eto compared to the latter alone. Due to this very favorable microbiological profile, AlpE has the potential to become an essential drug for the treatment of INH mono-resistant and M(X)DR TB as an alternative to fluoroquinolones or linezolid, thereby fostering antibiotic stewardship.

The reason why Eto is used in the majority of programs instead of Pto is poorly documented. The choice between using Eto or Pto can vary based on several factors, including regional availability, cost, side effect profiles, and historical treatment practices[48]. Alpibectir increased the susceptibility of *M. tuberculosis* to both Eto and Pto, equivalently.

In vivo in humans, an important fraction of Eto and Pto is metabolized to a sulfoxide derivative (Eto-SO and Pto-SO) by the host flavin-monooxygenases FMO1, FMO2.1, and FMO3[49]. These metabolites are known to be as potent as their progenitor against *M. tuberculosis* in vitro[49,50]. Because of a genetic polymorphism, Europeans and Asians lack FMO2.1. However, in sub-Saharan Africa, a region in which TB is a major health problem, a substantial proportion of individuals express FMO2.1. In contrast to FMO1 and FMO3, which are expressed primarily in the kidneys and liver, respectively, the main site of expression of FMO2 is the lungs. Thus, it cannot be excluded that these polymorphisms, as well as individual differences, might lead to variations in the transformation of Eto and Pto to their corresponding sulfoxides. It was thus essential to better understand how such possible differences could produce possible discrepancies in the boosting efficacy of alpibectir. Strikingly, we show here that alpibectir not only increases the potency of Eto and Pto, but also of their sulfoxide metabolites. This suggests that the final activity of AlpE or alpibectir/prothionamide is independent of FMO polymorphism as thioamides and their metabolites are potentiated equivalently.

Alpibectir was shown to be safe and well tolerated in a first-in-human study with healthy participants[22]. Alpibectir was recently investigated in a Phase 2a trial in South Africa; an open-label, randomized, 2-stage clinical trial that recruited adults with newly diagnosed, rifampicin- and isoniazid-susceptible pulmonary TB to determine bactericidal activity, pharmacokinetics, safety, and tolerability of Eto and alpibectir at various doses (NCT05473195). This trial established proof of concept, demonstrating that 27 mg of alpibectir potentiated the bactericidal activity of 250 mg of ethionamide to levels comparable to both the standard 750-mg ethionamide dose and 300 mg of isoniazid, while maintaining a favorable safety profile. These findings position the alpibectir-ethionamide combination as a promising candidate for future TB treatment regimens[23]. Due to the novel mode of action, this approach has the potential to create a paradigm shift in treatment options for antimicrobial resistance.

## Methods

### Reagents and consumables

Alpibectir batch RL01L234A0 was provided by BioVersys AG. Details of the synthesis of alpibectir and its analogue will be described elsewhere (manuscript in preparation). Eto (E6005), Pto (SMB00387), INH (I0500000), and DMSO (D8418) were purchased from Sigma.

### Chemical synthesis of Eto-SO and Pto-SO

Eto-SO and Pto-SO were synthesized as follows. Firstly, meta-chloroperoxybenzoic acid (2.97 g, 12.0 mmol for Eto-SO or 2.74 g, 11.1 mmol for Pto-SO) was added to a solution of Eto (2 g, 12.0 mmol) or Pto (2 g, 11.1 mmol) in MeOH (30 mL) at 0 °C. After 4 h of agitation at 0 °C, the solvent was removed *in vacuo*, and the crude product was purified by column chromatography (mobile phase: ethyl acetate/MeOH 90/10). The desired product was isolated as yellow microcrystals (yield: 0.65 g for Eto-SO and 1.41 g for Pto-SO). Eto-SO $^1$H NMR (300 MHz, DMSO-d6) δ (ppm) 9.38 (s, 1H), 8.56 (dd, J = 5.2, 0.5 Hz, 1H), 8.48 (s, 1H), 7.39 (s, 1H), 7.31 (dd, J = 5.2, 1.7 Hz, 1H), 2.78 (q, J = 7.6 Hz, 2H), 1.23 (t, J = 7.6 Hz, 3H). Pto-SO $^1$H NMR (300 MHz, DMSO-d6) δ (ppm) 9.39 (brs, 1H), 8.57 (dd, J = 5.2, 0.6 Hz, 1H), 8.50 (brs, 1H), 7.38 (s, 1H), 7.32 (dd, J = 5.2, 1.7 Hz, 1H), 2.74 (t, J = 7.5 Hz, 2H), 1.70 (s, J = 7.4 Hz, 2H), 0.91 (t, J = 7.4 Hz, 3H).

### Bacterial strains

All strains were cultured from frozen stocks in Middlebrook 7H9 medium (BD Difco) supplemented with glycerol 0.2% (Euromedex), Tween 80 0.05% (Sigma-Aldrich) and oleic acid-albumin-dextrose-catalase 10% (OADC, BD Difco) except if otherwise indicated.

**Table 3 | *M. tuberculosis* strains used in the study**

| Strain | Description |
| --- | --- |
| H37Rv | Reference WT genome |
| E1 | W4 Beijing *ethA* missense mutant (Gly343Ala) |
| 8 F | H37Rv *virS* missense mutant (Cys278Arg) |
| Rv3854::Tn | H37Rv *ethA* transposon insertion mutant (4327379) |
| Rv3083::Tn | H37Rv *mymA* transposon insertion mutant (3448753) |
| CD6R1 | H37Rv *ethA* frameshift (cAA302) and *mymA* deletion mutant (Δ13 kb) |
| H37Rv pMV261ΩinhA | H37Rv overexpressing InhA |
| H37Rv pMV261 | H37Rv transformed with empty pMV261 plasmid |
| H37Rv Δ*adhD* | Allelic replacement of *adhD* by hygromycin resistance gene *hygR* in H37Rv |

All strains used are briefly described in Table 3. Recombinant *M. tuberculosis* strains expressing an enhanced green fluorescent protein (GFP) were obtained by transformation with an integrative plasmid containing the *Aequoria Victoria egfp* gene[51]. Luciferase-expressing *M. tuberculosis* H37Rv was obtained by transformation with the pEG200 plasmid[52]. Except for strain E1, which is a derivative of the W4 Beijing strain, all other strains are derived from H37Rv[20]. Strains 8 F and CD6R1 are spontaneous mutants selected in vitro and Rv3854::Tn and Rv3083::Tn are transposon mutants[27]. CD6R1, Rv3854::Tn, and Rv3083::Tn were kindly gifted by Dr. Deborah Hung (Broad Institute, USA)[27]. *M. tuberculosis* H37Rv overexpressing InhA was obtained by transformation with plasmid pMV261ΩinhA[53]. As a control, H37Rv was also transformed with the empty pMV261 vector.

H37Rv Δ*adhD* was constructed as previously described with some modifications[54]. The DNA substrate for allelic replacement of *adhD* (Rv3086) consisted of the 500 bp upstream and downstream regions of *adhD* on either side of a cassette composed of *hygR* (hygromycin resistance gene) under its own promoter and the *sigA* promoter in frame with Rv3087 in order to restore the expression of the last genes of the operon. The construct was synthesized and cloned into a pUC57 vector between NdeI and HindIII restriction sites (GeneCust). *E. coli* DH5α strain transformed with the plasmid was grown in LB medium supplemented with 100 μg/mL ampicillin at 37 °C. The pUCΔ*adhD* plasmid was isolated (QIAprep Spin Miniprep kit, Qiagen), digested with NdeI and HindIII (FastDigest, Thermo Fisher Scientific), and gel purified (QIAquick gel extraction, Qiagen). *M. tuberculosis* H37Rv pJV53 was cultured in 20 mL 7H9 supplemented with 0.2% succinate and 20 μg/mL kanamycin in 25 cm² tissue-culture flasks. Cultures were incubated at 37 °C until mid-log phase and induced with 0.2% acetamide. Electrocompetent *M. tuberculosis* (200 μL) was transformed with 100 ng of salt-free dsDNA. Electroporated bacteria were resuspended in 6 mL 7H9 and following an incubation at 37 °C for 72 h, harvested by centrifugation at 3000 x *g* for 5 min and plated on 7H10 with 50 μg/mL hygromycin. Allelic exchange was confirmed by sequencing.

**Extracellular antibacterial activity**

Stock solutions of Eto, Pto, and INH were prepared in DMSO at 10 mg/mL, while alpibectir and its analogue were prepared at 10 mM. Aliquots were stored at −20 °C. To set up the assay plates, all compounds were initially transferred to a 384-well low-volume polypropylene plate (Corning 3672).

For checkerboard assays, ten 3-fold serial dilutions of Eto and alpibectir (or its analogue) were distributed vertically and horizontally, respectively, in black Greiner 384-well clear bottom polystyrene plates (Greiner 781091) using an Echo 550 liquid handler (Labcyte). The assay was conducted with two technical duplicates per biological replicate, with a total of three biological replicates. DMSO was added to relevant wells to ensure a final concentration of 0.6% across all wells. The assay

plates were inoculated with 50 μL/well of *M. tuberculosis* H37Rv cultures expressing GFP at an optical density ($OD_{600\ nm}$) of 0.02 and incubated at 37 °C for 5 days. GFP fluorescence ($\lambda_{ex} = 485\ nm$/$\lambda_{em} = 535\ nm$) was read using the Ensight Multimode plate reader (PerkinElmer, USA). For OD measurement of non-GFP expressing strains, an inoculum of $OD_{600\ nm}$ 0.002 was used and plates were incubated at 37 °C for 7 days before OD measurement at 600 nm using the Ensight multimode plate reader. $IC_{50}$ and $IC_{90}$ values were calculated by fitting fluorescence values for each concentration on GraphPad Prism version 10.2.2 (GraphPad Software, Inc., San Diego, CA).

For assays evaluating the antibacterial activity of Eto, Eto-SO, Pto, Pto-SO, or INH in combination with a fixed concentration of alpibectir, ten 3-fold serial dilutions of Eto, Eto-SO, Pto, Pto-SO, or INH were performed in duplicates into black Greiner 384-well clear bottom polystyrene plates (Greiner 781091) using an Echo 550 liquid handler (Labcyte). DMSO was added to relevant wells to ensure a concentration of 0.3% across all wells. To evaluate the effect of alpibectir on the antibacterial activity of the compounds, alpibectir was added at the indicated concentrations using the Echo 550 liquid handler. The assay plates were inoculated and incubated as mentioned above. GFP fluorescence or OD were measured and analyzed as detailed above.

**Intracellular antibacterial activity**

THP-1 cells were cultured at a density of $2 \times 10^5$ to $1 \times 10^6$ cells/mL in RPMI medium (Thermo Fisher 61870036) supplemented with 10% fetal bovine serum (FBS, Gibco 10500) at 37 °C, 5% $CO_2$.

For the intracellular assay in Fig. 1F, which measures luminescence of intracellular *M. tuberculosis*: A single-cell bacterial suspension of *M. tuberculosis* H37Rv pEG200 (expressing a luciferase operon) was prepared for infection. Briefly, an exponential phase culture was centrifuged at 3200 x *g* for 5 min and the pellet resuspended in RPMI-FBS before centrifugation at 100 x *g* to remove clumps. The supernatant was passed 5 times through a 0.9 mm diameter needle and the OD was measured. THP-1 cells (5 mL of $5 \times 10^6$ cells) were simultaneously infected by adding the bacterial suspension at a multiplicity of infection of 10 and differentiated with 40 ng/mL phorbol 12-myristate-13-acetate (Sigma P1585). The cells were incubated for 4 h at 37 °C, 5% $CO_2$ and the adherent cell monolayer was detached with a cell scraper. The cells were washed 4 times by centrifugation at 300 x *g* and resuspended in 25 mL of RPMI-FBS. Opaque white 384-well assay plates were prepared as described above for the extracellular assay. Plates were inoculated with 50 μL of the cell suspension ($1 \times 10^4$ cells/well) and incubated for 5 days at 37 °C, 5% $CO_2$. The antibacterial activity of the compounds was determined by measuring luminescence of intracellular bacteria. The experiment was performed in triplicate. For this assay, THP-1 cells transfected with Incucyte® CytoLight Red Lentivirus (EF-1 Alpha promoter, puromycin selection, Sartorius 4482) were used to enable measurement of their viability in the presence of drug treatment. The infected THP-1 cells (as described above) were distributed to black 384-well assay plates and incubated for 5 days. Images (3 fields per well) were acquired at 20X using the GE IN Cell 6500HS (GE Healthcare Life Sciences) automated confocal microscope with a 561 nm laser and 600/54 nm emission filters for detection of THP-1 cells. Images were processed using the Columbus image analysis software version 2.9.1 (PerkinElmer).

For the intracellular assay in Supplementary Fig. 4 of the Supplementary Information, which measures macrophage survival: Prior to infection, 5 mL of $5 \times 10^6$ THP-1 cells ($1 \times 10^6$ cells/mL) were differentiated with 40 ng/mL phorbol 12-myristate-13-acetate (Sigma P1585) and incubated for 3 days. A single-cell bacterial suspension was prepared for infection. Briefly, an exponential phase *M. tuberculosis* H37Rv culture was centrifuged at 3200 x *g* for 5 min and the pellet resuspended in RPMI-FBS before centrifugation at 100 x *g* to remove clumps. The supernatant was passed 5 times through a 0.9 mm diameter needle and the OD was measured. The medium of the adherent

THP-1 was renewed, and the bacterial suspension was added at a multiplicity of infection of 10. The cells were incubated for 2 h and then washed 3 times with RPMI. They were incubated with 50 mg/L amikacin (Sigma A1774) for 1 h to kill extracellular bacteria. The cell monolayer was washed 3 times with phosphate buffered saline (PBS, Sigma #D8537) and detached with trypsin-EDTA 0.25% (Gibco 25200056). The cell suspension was centrifuged at $100 \times g$ for 5 min and the pellet resuspended in 25 mL of RPMI-FBS. 384-well assay plates were prepared as described above for the extracellular assay. Plates were inoculated with 50 μL of the cell suspension ($1 \times 10^4$ cells/well) and incubated for 5 days at 37 °C, 5% $CO_2$. The antibacterial activity of the compounds was determined by measuring macrophage survival. On day 4, 5 μL of 0.05% resazurin was added to each well and resorufin fluorescence was read on day 5 after an incubation of 16 h at 37 °C, 5% $CO_2$. The experiment was performed in triplicate.

### Time-kill assay

Serial ten-fold dilutions of Eto, INH, and alpibectir were prepared in DMSO. Appropriate volumes for each treatment were added to non-ventilated 25 cm² tissue-culture flasks. DMSO volume was compensated so that the concentration of DMSO in all flasks was equal to 0.25%.

The flasks were inoculated with 10 mL of *M. tuberculosis* H37Rv WT or Δ*adhD* cultures diluted to an $OD_{600\ nm}$ of 0.005 (theoretically corresponding to $5 \times 10^5$ CFU/mL) and incubated at 37 °C without shaking.

Cultures from each flask were plated on square plates containing Middlebrook 7H10 solid medium (BD) supplemented with glycerol 0.2% (Euromedex), Tween 80 0.05% (Sigma-Aldrich) and oleic acid-albumin-dextrose-catalase 10% (OADC, BD Difco) on days 0, 2, 4, 7, 14, and 21. On day 0, only 3 cultures were plated whereas on other days, all cultures were plated. Before plating, cultures were 10-fold serially diluted (up to $10^{-7}$) and 10 μL of each serial dilution and of the non-diluted culture were plated. CFUs were counted after incubating the plates for 3 weeks at 37 °C. The lowest possible number of countable CFUs (limit of detection) is 1 when plating 10 μL of a non-diluted culture, which amounts to a $\log_{10}$ CFU/mL of 2.

Three biological replicates were performed for H37Rv WT and three replicates for H37Rv Δ*adhD*.

### Transcriptomics

*M. tuberculosis* H37Rv cultures were diluted to an $OD_{600\ nm}$ of 0.2 from a fresh exponentially growing culture and were incubated at 37 °C without shaking until an $OD_{600\ nm}$ of 0.7 was reached. To study the effect of alpibectir alone on gene expression levels, 25 mL cultures were treated with 0.1 mg/L of alpibectir for 24 h in triplicate. To study the effect of Eto combined with alpibectir, cultures were first treated for 20 h with 0.1 mg/L alpibectir followed by an additional 4 h with 2 mg/L Eto. The final DMSO concentration in all cultures was equal to 0.1%. Cultures were treated only with 0.1% DMSO as controls (in triplicate for comparison with alpibectir alone or in a single replicate for comparison with alpibectir and Eto treatment). RNA extraction and transcriptomics analysis were performed as previously described[21]. Briefly, after the treatment duration, 10 mL of the cultures were harvested by centrifugation at $3000 \times g$ for 5 min, resuspended in 1 mL of RNApro™ (FastRNA Pro Blue Kit, MP biomedicals) and homogenized in impact-resistant 2 mL tubes containing 0.1 mm silica spheres (Lysing Matrix B, MP biomedicals) using a FastPrep FP120 cell disrupter (Thermo Fisher Scientific) at 6.0 Hz for 40 s. The ribolysed cells were centrifuged at $12,000 \times g$ to remove cellular debris and RNA was purified following the manufacturer's instructions. Ribosomal RNA (rRNA) depletion was performed using QIAseq FastSelect −5S/16S/23S Kits; Qiagen. Libraries for Illumina sequencing were prepared with the TruSeq RNA sample preparation kit version 2.0 rev. A (Illumina Inc.). All cDNA libraries were uniquely indexed. cDNA libraries were

sequenced using an Illumina NextSeq 500 system (Illumina Inc.) in high-output mode. All samples were multiplexed on one lane of the flow cell and sequenced in single-read sequencing mode with read lengths of 150 bp. Raw RNA-seq reads were processed with Illumina quality control tools using default settings. Sequences shorter than 50 bp and/or that contained any 'Ns' and/or with a mean quality score lower than 30 were removed using PRINSEQ. Next, rRNA-specific reads were filtered out by mapping all the reads on *M. tuberculosis* rRNA sequences using Bowtie2 (http://bowtie-bio.sourceforge.net/bowtie2/index.shtml). Analysis of the RNA sequencing data was conducted using the SPARTA open-source software package with default parameters (https://sparta.readthedocs.io/en/latest/). A total of 4036 genes could be analyzed. Raw and processed RNA-seq reads were deposited in the ArrayExpress repository (accession code E-MTAB-16253).

### Proteomics

Protein extraction and proteomics analysis were performed as previously described[21]. *M. tuberculosis* H37Rv was cultured in 7H9 without glycerol and supplemented with 2% (w/v) glucose and 0.025% (v/v) tyloxapol at 37 °C for about 7 days until an $OD_{600\ nm}$ of 0.8. The culture was diluted to an $OD_{600\ nm}$ of 0.2 and split in 4 subcultures. Two of them were further incubated for 72 h in the presence of 1 μM (0.33 mg/L) alpibectir and the others were similarly incubated with DMSO. Protein extracts were prepared by centrifuging the culture and washing the pellet with PBS. The pellet was resuspended in 1 mL lysis buffer (50 mM Tris-HCl, pH 7.4, 0.8 (v/v) % NP-40, 1.5 mM $MgCl_2$, 5% glycerol, 150 mM NaCl, 25 mM NaF, 1 mM $Na_3VO_4$, 1 mM dithiothreitol (DTT) and one Complete EDTA-free protease inhibitor tablet (Roche)) and disrupted using a TissueLyser II (Qiagen) for three cycles at full amplitude for 5 min in refrigerated supports. Bacterial lysates were centrifuged at $14,000 \times g$ for 30 min and the supernatant filtered using Millex LG (PTFE) 0.2 μm/13 mm diameter. Filtered supernatants were further ultracentrifuged at 4 °C for 60 min ($140,000 \times g$, 55,000 rpm). Samples were prepared in quadruplicate.

SDS sample buffer (2% SDS, 10% glycerol, 0.005% bromophenol blue, 100 mM Tris-HCl, 125 mM Tris Base) was added to 37.5 μL of the *M. tuberculosis* protein extract. Proteins were digested according to a modified single pot solid-phase sample preparation (SP3) protocol[55,56]. Briefly, proteins were bound to paramagnetic beads (SeraMag Speed beads, GE Healthcare, CAT#45152105050250, CAT#651521050502) in 50% ethanol (final concentration). Beads were washed 4 times with 70% ethanol and proteins digested by resuspending in 0.1 mM HEPES (pH 8.5) containing TCEP, chloroacetamide, trypsin, and LysC followed by overnight incubation.

Peptides were labeled with isobaric mass tags (TMT10, Thermo Fisher Scientific, Waltham, MA) using the 10-plex TMT reagents, enabling relative quantification of 10 conditions in a single experiment[56,57]. The labeling reaction was performed in 40 mM triethylammoniumbicarbonate, pH 8.5 at 22 °C and quenched with glycine. Labeled peptide extracts were combined into a single sample per experiment, lyophilized and subjected to LC-MS analysis.

Samples were pre-fractionated by reversed-phase chromatography at high pH to 16 samples of which 10 were measured[58].

After lyophilization, dried samples were resuspended in 0.05 % trifluoroacetic acid in water. Half of the sample was injected into an Ultimate3000 nanoRLSC (Dionex) coupled to a Q-Exactive (Thermo Fisher Scientific). Peptides were separated into custom-made 35 cm × 100 μm (ID) reversed-phase columns (Reprosil) at 55 °C, gradient elution was performed from 3.5% acetonitrile to 29% acetonitrile in 0.1% formic acid and 3.5% DMSO over 120 min.

The Q Exactive Plus operated with a data-dependent top 10 method. MS spectra were acquired using 70.000 resolution and an ion target of 3E6. Higher energy collisional dissociation (HCD) scans were performed with 35% NCE at 35.000 resolution (at m/z 200), and ion

target settings was set to 2E5 so as to avoid coalescence[56]. The instruments were operated with Tune 2.4 and Xcalibur 3.0 build 63.

Mascot 2.4 (Matrix Science, Boston, MA) was used for protein identification with a 10 parts per million mass tolerance for peptide precursors and 20 mD (HCD) mass tolerance for fragment ions.

To create the fasta file for mascot searching, all proteins corresponding to the taxonomy 'Mycobacterium tuberculosis H37Rv' were downloaded from Uniprot (release 20170621) and supplemented with common contaminant protein sequences of bovine serum albumin, porcine trypsin and mouse, rat, sheep and dog keratins. To assess the false discovery rate (FDR), "decoy" proteins (reverse of all protein sequences) were created and added to the database, resulting in a database containing a total of 7,994 protein sequences, 50% forward, 50% reverse.

Unless stated otherwise, we accepted protein identifications as follows: (i) For single spectrum to sequence assignments, we required this assignment to be the best match and a minimum Mascot score of 31 and a 10x difference of this assignment over the next best assignment. Based on these criteria, the decoy search results indicated < 1% false discovery rate (FDR). (ii) For multiple spectra to sequence assignments and using the same parameters, the decoy search results indicated < 0.1% FDR. Quantified proteins were required to contain at least 2 unique peptide matches. FDR for quantified proteins was < 0.1%.

Protein abundance ratios were calculated by IsobarQuant[59]. Fold-change ratios were calculated from sum ion areas of compound-treated samples and compared to one vehicle control that was set to 1. The complete list of analyzed proteins is provided as Supplementary Data 1.

## In-cell transcriptional regulator assay

The in-mammalian cell bacterial regulator reporter assay was performed as previously described in refs. [20,21]. To construct chimeric bacterial-mammalian transcriptional regulators, the coding sequence of EthR, EthR2, or VirS was fused to the Herpes simplex derived transactivator protein VP16. The sequence was optimized for expression in human/mouse, synthesized (Genscript) and introduced into pSEAP2-control (Clontech) using EcoRI/XbaI to generate expression vectors $P_{SV40}$-EthR-VP16-HA-pA (pCK310), $P_{SV40}$-EthR2-VP16-HA-pA (pVT20), and $P_{SV40}$-VirS-VP16-HA-pA (pCK461), respectively. To enable transcriptional control by the chimeric transcriptional regulators constructed above, the promoter region recognized by EthR, EthR2, or VirS, i.e., the intergenic region between ethA and ethR, between ethA2 and ethR2, or between virS and mymA, was cloned upstream of a minimal variant of the human cytomegalovirus-derived promoter ($P_{hCMVmin}$) into pSEAP2-basic (Clontech) using XhoI/EcoRI to generate $O_{EthR}$-$P_{hCMVmin}$-SEAP-pA (pCK311), $O_{EthR2}$-$P_{hCMVmin}$-SEAP-pA (pVT23), $O_{VirS}$-$P_{hCMVmin}$-SEAP-pA (pCK462), respectively.

Baby hamster kidney cells (BHK-21, American Type Culture Collection CCL-10) were cultured in Dulbecco's modified Eagle's medium (DMEM, Gibco 41966) supplemented with 10% (v/v) heat-inactivated FBS and 1% (v/v) penicillin/streptomycin (Gibco 15140) in a humidified atmosphere with 5% $CO_2$ at 37 °C. Prior to transfection, cells from a confluent grown culture dish were split 1:2 to a new petri dish and grown for 16 hours. For transfections with the EthR and EthR2 regulatory systems, 10 μg of total plasmid DNA (8 μg pCK310 and 2 μg pCK311 or 5 μg pVT20 and 5 μg pVT23) were added to 1 mL Opti-MEM reduced serum medium (Gibco 11058021), mixed with 30 μL MegaTran 1.0 transfection reagent (Origene TT200003) and incubated for 15 min at room temperature. For transfections with the VirS regulatory system, 10 μg of total plasmid DNA (5 μg pCK461 and 5 μg pCK462, ratio 1:1) were mixed with 0.5 mL 1x Jetprime buffer and 10 μL of jetPRIME transfection reagent (Poly Transfection 114-15), vortexed for 10 sec, and incubated for 10 min at room temperature. The DNA mixes were each transferred to the cells and incubated for 6 h for plasmid uptake. The medium was then removed, the cells washed twice with PBS and detached by adding 1 mL of 0.05% trypsin-EDTA (Gibco 25300). The

cells were resuspended in 8 mL DMEM to neutralize the trypsin, counted, and diluted to 500,000 cells/mL using fresh culture medium before 100 μL of the suspension was distributed per well in a 96-well plate. Compound dilutions were prepared by adding 4.8 μL of alpibectir (10 mM stock solution in DMSO) to 600 μL culture medium. For EthR and EthR2, 100 μL of a two-fold dilution of the compound solution were added to the cells to reach final compound concentrations of 20 nM to 40 μM ( = 13.3 mg/L). For VirS, 100 μL of a five-fold dilution of the compound solution were added to the cells to reach final compound concentrations of 0.0008 nM to 40 μM ( = 13.3 mg/L). After incubation for 48 h, SEAP expression was quantified from cell culture supernatants using the p-nitrophenyl phosphate-based method[57].

## Binding interaction between alpibectir and VirS

The recombinant protein VirS was produced as previously described in ref. [59]. Briefly, BL21(DE3) cells transformed with pET28aΩVirS were cultured in LB medium containing 50 μg/mL kanamycin and protein expression was induced with 1 mM isopropyl-β-D-thiogalactoside overnight at 28 °C. Harvested bacteria were suspended in lysis buffer (1X PBS, 100 mM L-arginine, 5 mM imidazole) containing a tablet of cOmplete EDTA-free protease inhibitor cocktail (Roche) and lysed by Emulsiflex-C3 (Avestin). The lysate was centrifuged (46,000 x g, 4 °C) and the supernatant incubated with HisPur Cobalt-NTA affinity resin (Thermo Fisher Scientific) at 4 °C for 1 h. VirS was eluted in 1X PBS, 100 mM L-arginine, 1 M NaCl, 500 mM imidazole, and 10% glycerol. The binding interaction between alpibectir and VirS was evaluated by thermal shift assay (TSA), conducted in an iCycler RT-PCR system (Bio-Rad, Hercules, CA). SYPRO Orange (Thermo Fisher Scientific) was used as extrinsic fluorescent dye at a final concentration of 5X and VirS was used at a final concentration of 8.17 mg/L. Alpibectir was added at a concentration comprised between 0.08 and 33.3 mg/L. Sample solutions were dispensed into a 96-well optical reaction plate (MicroAmp™ Optical, Thermo Fisher Scientific) and the plate was sealed with optical PCR plate sheet (Thermo Fisher Scientific). The final DMSO concentration was equal to 0.5% in all tested conditions. The plate was heated from 20 to 95 °C with a heating rate of 0.7 °C/min. The fluorescence intensity was recorded with wavelength ranges of 450–490 and 560–580 nm for excitation and emission, respectively. Each condition was measured at least in quintuplicate. Data were analyzed with GraphPad Prism 10 (GraphPad Software, Inc., San Diego, CA).

Crystallization and structure determination of the alpibectir/VirS complex was performed as previously described in ref. [59]. Briefly, a VirS construct (6HIS-SET3a-VirS) integrated in the pET28a plasmid was expressed recombinantly in E. coli and purified to homogeneity using cobalt-NTA affinity resin. Crystals of alpibectir/VirS complex were obtained through an in situ proteolysis approach. The method involved incubating a mixed solution of VirS and alpibectir (at molar ratio of 1:1) on ice for 20 min, followed by the addition of subtilisin (final concentration of 0.1 mg/mL). Subsequently, sitting-drop vapor diffusion crystallization experiments were conducted at 20 °C using a reagent solution containing 0.3 M sodium acetate trihydrate, 0.1 M Tris pH 7.5, 8% (w/v) PEG20,000, 8% (v/v) PEG500-MME. The crystal structure was solved by the molecular replacement method using the MOLREP program[59] and the VirS AlphaFold2 model[60] as search template. Data collection and refinement statistics are provided in SI Table 2. An electron density omit map was generated by excluding the alpibectir molecule from the structure, then randomly modifying the remaining atom coordinates within a range of 0–0.3 Å, followed by updating the corresponding structure factors and computing a residual map. The analysis of the interactions between alpibectir and VirS in the crystal structure was facilitated using the LigPlot+ program[61].

## Binding interaction between alpibectir and α-latalbumin

As a negative control, the binding interaction between alpibectir and the α-latalbumin protein (Sigma, L-5385) was evaluated by thermal

shift assay (TSA), conducted in a CFX-Opus 96 Dx RT-PCR detection system (Bio-Rad CFX-Opus 96, Hercules, CA). SYPRO Orange (Thermo Fisher Scientific) was used as an extrinsic fluorescent dye at a final concentration of 5X and $\alpha$-latalbumin was used at a final concentration of 5 $\mu$M. Concentrations of alpibectir ranged from 0.250 to 100 $\mu$M. Sample solutions were dispensed into a 96-well optical reaction plate (MicroAmp™ Optical, Thermo Fisher Scientific) and the plate was sealed with optical PCR plate sheet (Thermo Fisher Scientific). The final DMSO concentration was kept at 0.5% in all wells. The plate was heated from 20 to 85 °C with a heating rate of 1.8 °C/min. The fluorescence intensity was recorded with a range of excitation and emission wavelengths of 450-490 and 560-580 nm, respectively. Each condition was measured at least in quintuplicate. Data was analyzed with Prism 10 (GraphPad Software Inc.).

### Determination of the FoR to alpibectir/Eto
Resistant mutants were selected on 7H11 plates containing various drug concentrations. Plates were prepared in triplicate to spread 1 mL of culture containing $10^7$, $10^8$, $10^9$ bacteria. The inoculum was prepared by sonicating an exponentially growing culture of *M. tuberculosis* H37Rv using a LabSonic M (Sartorius) sonicator equipped with a Microtip MS1 of 1 mm diameter for 1 min at 80% amplitude and a 0.8 cycle. Colonies were counted after an incubation of 21 days at 37 °C.

To test for cross-resistance, 14 clones selected on agar plates containing alpibectir/Eto combinations (and cultured with selective pressure) were streaked on plates containing either no antibiotic or moxifloxacin (1 mg/L; GSK978567A), linezolid (1 mg/L; GSK GW602383X), ethambutol (4 mg/L; GSK GF133424A), bedaquiline (0.5 mg/L; GSK3377910A), pretomanid (PA824, 2 mg/L; GSK3184596A), or rifampicin (0.5 mg/L; Sigma R3501). These concentrations correspond to 2x the MIC or, whenever available, 2x the epidemiological cut-off (ECOFF; representing measures of a drug MIC distribution that separate bacterial populations into those representative of a WT population and those with acquired or mutational resistance to the drug).

### Determination of MIC values by Bactec™ MGIT™ 960
Minimal inhibitory concentration (MIC) values of compounds against *M. tuberculosis* H37Rv or clinical strains were determined in Bactec™ MGIT™ 960 culture tubes supplemented with 10% OADC and inoculated with an exponential phase preculture of the corresponding bacterial strain. Tubes were continuously monitored and analyzed using the EpiCenter software, thus enabling a precise assessment of bacterial growth. The minimal inhibitory concentration is defined as "no growth detected" at the time a 100-fold diluted control culture (Growth control 1%) reaches 400 Growth Unit. An extended incubation period of a further 7 days following the positivity of the "Growth control 1%" allows determination of the drug concentration at which no bacterial "late growth" is observed.

Eto was tested at 2, 5, and 10 mg/L and in a few instances also at 20 mg/L. Alpibectir was tested alone at 0.05 mg/L. Alpibectir at 0.017 mg/L and in a few cases 0.05 mg/L was tested in the presence of 0.8 mg/L Eto. Growth was measured by detection of oxygen consumption in the BD BACTEC™ MGIT™ 960 automated system in all tubes when the control tube showed growth.

### Mouse efficacy study
To assess the bactericidal activity of boosted Eto, we used a 4-week treatment model of TB. A total of 175 4-week-old female Balb/c mice were intravenously infected into the tail vein with 0.5 mL of a calibrated bacterial suspension and an injected bacillary load of $10^6$ CFU at D0[62].

Eto, INH, and alpibectir were suspended in 1% methylcellulose (MC) (400cPs) in water. Alpibectir/Eto combinations were formulated together into 1% MC. Mice were randomly allocated to 20 groups. A pretreatment group consisted of 12 mice that were sacrificed at D1. A negative control group consisted of 12 infected mice that were not treated. Treatment was started the day after infection and given orally on 6 out of 7 days for 4 weeks (D28). The treatment groups consisted of 6 or 12 mice each; for the latter, 6 mice were sacrificed after 2 weeks (D14) of treatment. A positive control group was treated with INH alone at the human equivalent dose of 25 mg/kg[63]. Eto (5, 15, 50, 100, or 200 mg/kg) and alpibectir (0.1, 0.5, or 1.6 mg/kg) were either given as monotherapy or Eto (5, 15, or 50 mg/kg) was combined with alpibectir (0.1, 0.5, or 1.6 mg/kg). Treatment efficacy was assessed in survival rate at D28 and bacillary load reduction at D14 and D28 compared to pretreatment (D1).

Mice were euthanized by cervical elongation. Lungs were removed, homogenized, serially diluted, and spotted onto drug-free Lowenstein Jensen for the determination of colony forming units (CFUs).

CFU numbers were converted to CFU/lung by multiplying the number of CFUs by the volume of the lung homogenate spotted and the dilution at which the colonies were counted. CFU data was then transformed into $\log_{10}$ CFU/lung for calculation of means and standard deviations (SD). The changes in the $\log_{10}$ CFU/lung at the endpoint compared to the pretreatment (D1) were calculated. Dose-effect curves were fitted to the $\log_{10}$D CFU data by using a sigmoidal maximum-effect ($E_{max}$) model (GraphPad Prims, version 10.1.2, GraphPad Software Inc, San Diego, CA, USA) to determine the effect of Eto or of alpibectir/Eto combinations on the change in CFUs. The total daily dose (TDD) needed for 1-$\log_{10}$ growth, stasis and 1-$\log_{10}$-kill were determined. Mean CFU counts were compared using one-way analysis of variance (One-Way ANOVA, no matching or pairing, assuming Gaussian distribution and equal SDs, correction for multiple comparisons with Dunnett test (recommended by Prism); $F\,(19, 103) = 35.59$, $p < 0.0001$, $R^2 = 0.8678$). The exact p-values are mentioned in the text. Data from D14 and D28 were analyzed separately. All experiments were performed in female mice; sex was not considered as a variable in the study design. All animals were grown in an animal facility under filtered air conditions (20–24 °C), with a 12 h light/dark cycle and relative humidity ranging from 45 to 85%, in plastic cages using sterilized wood shavings as bedding.

### Statistics & Reproducibility
All experiments were conducted with at least 2 independent biological replicates as detailed in figure legends and "Methods". No statistical methods were used to predetermine sample sizes, which are considered as standard. No data were excluded from experiments except the in vivo study for which some mice were excluded from analysis due to culture contamination as indicated in Supplementary Table 7A. Experiments were not randomized except the in vivo study for which mice were randomly assigned to treatment and control groups. The investigators were not blinded to allocation during experiments and outcome assessment. Statistical analyses were carried out using GraphPad Prism version 10.2.2 (GraphPad Software Inc.). Data are presented as mean ± SD when applicable. For the in vivo study, statistical significance was analyzed using one-way ANOVA.

### Ethical statement
The in vivo experimental project was favorably evaluated by the Charles Darwin Ethics Committee no. 005, located at Pitié-Salpêtrière Hospital (France), and clearance was given by the French Ministry of Education and Research under the APAFIS no. 12380-2017112809414820 v3. The animal facility was authorized to conduct animal experiments (license number C-75-13-08). The individuals involved in the animal experiments underwent specific training recognized by the French Ministry of Education and Research.

## Reporting summary

Further information on research design is available in the Nature Portfolio Reporting Summary linked to this article.

## Data availability

The electron density map and the coordinates of the refined crystal structure were deposited to the Protein Data Bank under PDB ID: pdb_00008RCX [https://www.rcsb.org/structure/8RCX]. Transcriptomics data were deposited in the ArrayExpress repository with the accession code E-MTAB-16253. The mass spectrometry proteomics data were deposited to the ProteomeXchange Consortium via the PRIDE partner repository with the dataset identifier PXD075165. Source data are provided with this paper.

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

## Acknowledgements

The PROXIMA1 and PROXIMA2 beamlines of the SOLEIL synchrotron are thanked for initial X-ray diffraction screens (BAG20210875). We are grateful to Matthew Bowler for access to the ESRF MASSIF-1 facility. We acknowledge the technical support of Dr. Florence Leroux, the *ARIADNE*-Screening (UAR2014 - US41-*PLBS*), and *ARIADNE*-ADME facilities (French national infrastructure ChemBioFrance). TRIC-TB has received funding from the Innovative Medicines Initiative 2 Joint Undertaking (JU) under grant agreement No 853800. The JU receives support from the European Union's Horizon 2020 research and innovation program and through the European Federation of Pharmaceutical Industries and Associations (EFPIA). This research was supported by Inserm, CNRS, the University of Lille, the French government's Investissements d'Avenir program (grant reference: Mustart ANR-20-PAMR-0005), and the INTHREPIDE-Antibioresistance program (Région Hauts de France Convention N°2108838). The Belgian National Reference Center for Mycobacteria is partially supported by the Belgian Ministry of Social Affairs through funding within the Health Insurance System. Thomas Maitre was funded by INSERM as part of the "Poste Accueil INSERM" program and by the Fondation du Souffle as part of the "Formation pour la Recherche 2020" program.

## Author contributions

A.R.B., M.P., N.W., and M.J.R. conceived and designed the study. N.W., B.D., and M.J.R. designed and coordinated synthesis of chemical compounds. T.M. and N.V. designed and coordinated the animal study. R.W. designed and coordinated the biochemical and structural characterization of VirS/alpibectir binding. S.L., G.E.D., L.B., D.B.-A., and M.G. supervised the overall research program. Z.E., C.G., T.M., and R.F. performed the studies. A.C., L.F.L.R., A.G., A.A., R.A., L.T., S.S., V.T., B.S., L.H., C.K., F.J., E.P.H., A.M., M.J.R.L., S.G.D., T.W., V. Mathys, K.S., and V. Megalizzi participated in data collection. A.D. and M.B. synthesized the sulfoxide derivatives of Eto and Pto. Z.E., C.G., T.M., R.F., S.L., R.W., N.V., A.R.B., M.P., N.W., and M.J.R. were involved in data analysis. Z.E., C.G., T.M., N.V., N.W., M.P., and A.R.B. wrote the manuscript. All authors reviewed drafts of the manuscript and gave final approval to submit for publication.

## Competing interests

M.B., V.T., B.S., A.D., L.H., C.K., S.L., G.E.D., F.J., M.G., and M.P. are current or former BioVersys employees and hold stocks or stock options in the company. Z.E and R.F. are former BioVersys employees. The Sorbonne Université Team (A.C., L.F., Th.M., A.G., N.V., A.A.) has received a research grant from BioVersys for conducting preclinical studies on Alpibectir/Ethionamide, and is involved in the Respiri-NTM IMI-AMR project in collaboration with BioVersys. T.W. and S.G.D. are current Cellzome GmbH (a GSK company) employees and hold stocks or stock options in GSK. D.B-A., M.J.R., A-M.L., and L.B. are current or former GSK employees and hold stocks or stock options in the company. M.J.R.L. is former GSK employee. N.W., B.D., M. J. R., E. P-H, and M.B. are listed as co-inventors on patent WO 2019/034700 Al entitled "Novel

compounds." N.W. and A.R.B. are consultants for BioVersys. B.D. is paid as scientific co-director of Institut Pasteur de Lille. A.R.B. team and the research unit U1177 - Drugs and Molecules for Living Systems received research funding from BioVersys. The remaining authors have no competing interest to declare.

## Additional information

[1]Univ. Lille, CNRS, Inserm, Institut Pasteur de Lille, U1019 - UMR 9017 - CIIL - Center for Infection and Immunity of Lille, Lille, France. [2]BioVersys SAS, Lille, France. [3]Research Department in Drug Development, Faculty of Pharmacy, Université Libre de Bruxelles, ULB, Bruxelles, Belgium. [4]Sorbonne Université, Inserm, Centre d'Immunologie et des Maladies Infectieuses, CIMI, F-75013 Paris, France. [5]APHP. Sorbonne Université, Hôpital Tenon, Service de Pneumologie et d'Oncologie Thoracique, Centre de Référence Maladie Rares, GRC Sorbonne Université SOLID, Paris, France. [6]APHP. Sorbonne Université, Hôpital Saint-Antoine, Département de Bactériologie, Centre National de Référence des Mycobactéries, Paris, France. [7]APHP. Sorbonne-Université, Hôpital Pitié Salpêtrière, Laboratoire de Bactériologie-Hygiène, Centre National de Référence des Mycobactéries et de la résistance des mycobactéries aux antituberculeux, Paris, France. [8]BioVersys AG, Basel, Switzerland. [9]GSK, Tres Cantos R&D, PTM, Tres Cantos, Madrid, Spain. [10]Cellzome GmbH. A GSK Company, Heidelberg, Germany. [11]National Reference Center for Tuberculosis and Mycobacteria, Sciensano, Brussels, Belgium. [12]Univ. Lille, Inserm, Institut Pasteur de Lille, U1177 - Drugs and Molecules for Living Systems, Lille, France. [13]Univ. Lille, CNRS, Inserm, CHU Lille, Institut Pasteur de Lille, US 41 - UMS 2014 - PLBS, Lille, France. [14]These authors contributed equally: Zainab Edoo, Camille Grosse, Thomas Maitre, Rosangela Frita. [15]These authors jointly supervised this work: Benoit Deprez, Modesto J Remuiñán, Nicolas Willand, Michel Pieren, Alain R. Baulard. ✉e-mail: nicolas.willand@univ-lille.fr; michel.pieren@bioversys.com; alain.baulard@pasteur-lille.fr

