## [Transparent Peer Review file · Nature Communications]

Alpibectir–Ethionamide combination (AlpE) for the treatment of tuberculosis

Corresponding Author: Dr Alain Baulard

Version 0:

Reviewer comments:

Reviewer #1

(Remarks to the Author)

The manuscript presents interesting findings on the potential of alpibectir to enhance the antibacterial activity of ethionamide (Eto) and prothionamide (Pto) through its interaction with the transcriptional regulator VirS. The study is well-designed, combining biophysical, genetic, and cellular assays to demonstrate the mechanism of action and therapeutic potential of alpibectir. However, some minor revisions are needed to strengthen the manuscript.

1. In Line 285, The author mentioned “In contrast to the intrinsic growth-inhibitory activity shown in Fig.4, no bactericidal effect was observed when alpibectir was used alone at concentrations up to 1 mg/mL” and attribute the results to the impact on bacterial adaptability. However, the animal assay of alpibectir showed a decrease of Mtb CFU when treat with 1.6 mg/mL of alpibectir. Can the author provide a more detailed discussion on this results?
2. In Line 258-262, the author suggested polar effect may occur during KO of adhD. The gene complementation experiment needs to be conducted.
3. In transcriptomic/proteomic analysis, the author only showed sharp LFC of selected genes. However, it is necessary to declared the statistically significance. Also, a volcano plot with annotation of these differentially expressed genes would also be more intuitive.
4. It would be beneficial to include an INH + alpibectir treatment group as an additional control in Fig. 6. This would help to distinguish whether the observed effects are unique to the Eto+alpibectir combination or not.
5. The authors presented a virS/mymA-inactivated Beijing strain in Table 3. Is this mutation frequently observed in clinical strains? This would determine the effectiveness of the combination therapy.

Reviewer #2

(Remarks to the Author)

The manuscript by Edo et al. reports work using a combination of orthogonal approaches to show that Alpibectir, an activator of VirS with bactericidal activity against Mycobacterium tuberculosis (Mtb), boosts the activation of the oxidoreductase MymA, promoting ethA-independent conversion of ethionamide into its active form to potentiate inhibition of InhA. The authors show that ethionamide administered in combination with Alpibectir led to lower bacterial loads in Mtb-infected mice. The work presented in the manuscript is well executed, robust and cogent. However, some aspects of the work should be clarified and discussed in more detail. Specific comments are as follows:

- It is unclear whether Alpibectir alone can kill intracellular Mtb. In Figure 1F, Alpibactir at concentrations ranging from 0.050-0.333mg/L does not prevent replication of intracellular Mtb at concentrations at which ethionamide is ineffective against Mtb, suggesting that Alpibectir is not effective against intracellular Mtb. This should be clearly discussed. In addition, the authors use a rather unusual murine model of tuberculosis wherein the drug is administered 24 hours post infection. Rationale for using this model should be included. Did the lack of effect of Alpibectir against intracellular Mtb motivate the use of this model?. Does the short time lapse between infection and treatment amplify the effect of Alpibectir alone because the drug

helps reduce the extracellular Mtb load?

- Figure 2, panel B. Although the data are compelling, some relevant controls are missing in the thermal denaturation experiment. The authors should either repeat it with inclusion of another protein that does not bind Alpibectir, or ideally with inclusion of a variant of VirS that cannot bind Alpibectir. The structural data should help in designing such a variant.
- Figure 4. There is no attempt to complement the Δ adhD mutant. It would be useful to complement the Δ adhD mutant with adhD and each of the three downstream genes individually first, and then in combination if unsuccessful, to define which gene(s) are responsible for the intrinsic activity of Alpibectir.
- Figure 5. The molecular mechanisms underlying ethionamide potentiation by Alpibectir remain unclear. Ethionamide is primarily converted into its active form by EthA in the absence of Alpibectir, the addition of which favours bioconversion of ethionamide by MymA. If EthA and MymA activate ethionamide into the same bioactive form (the NAD adduct), do they do so at the same rate? For the same concentration of ethionamide, how do levels of the bioconverted form compare in H37Rv versus H37Rv treated with Alpibectir? Does activation by MymA allow for larger accumulation of the bioconverted, active form compared to activation by EthA? This could explain why the combination of Alpibectir/ethionamide is better able to kill drug-recalcitrant Mtb subpopulations, and why the combination is effective at killing strains overexpressing InhA (lines 368-373). These measurements should be done; at the very least, these questions should be discussed.
- p12, line 282: the use of "only" might be misleading as ethionamide alone at 2.5 or 25 mg/L leads to statistically similar levels of bacterial clearance at day 7.
- The grounds on which the authors conclude that InhA is expressed at a basal level or overexpressed are unclear.
- How do the authors explain the absence of dose effect between administration of 15 and 50mg/kg of ethionamide?
- Please check the policy of Nature Communications regarding the use of "data not shown".

Reviewer #3

(Remarks to the Author)

The objective of discovering new treatments for tuberculosis is an undeniable priority, especially for those areas and regions and which tuberculosis have a high incidence for example in sub-Saharan Africa and in south-east Asia. The reporting of novel drugs or drug combinations are therefore not only important but necessary.

In this study Edoe and colleagues report on the novel drug combination that involves Alpibectir and ethionamide. This study reports on it to effects in studies that were directly conducted against *M. tuberculosis* bacilli, against bacilli harboured in macrophages as well as its effects tested in mice.

General Comments:

The study design is well constructed with logical flow in the experimental steps. The findings that alpibactir potentiates Eto activity and has antibacterial activity independent of Eto is highly promising.

In Fig.1A the effects of alpibactir is generally evaluated against the effect of Eto in its absence. While this is a necessary control, the inclusion of a compound in the same class of amino piperidines that does not potentiate Eto is an additional control required to demonstrate specificity.

In the next phase of the study, the effects of Alp/E was tested against intracellular bacilli (Fig 1F). The authors report efficacy, however whether macrophages remain intact and viable for the duration of the assay is not presented. Should macrophages be lysed, the test would essentially evaluate efficacy against extracellular bacilli. The inclusion of a data that demonstrate macrophage viability for the duration of the experiment must be included to validate the conclusion of efficacy against intracellular bacilli. Again, while the absence of alpibactir serves as a control in Fig 1F, a control to demonstrate drug specificity is absent.

In line 169, the authors refer to the "...exceptional specificity of alpibactir...". The inclusion of a non-specific control for alpibactir would enhance this claim. The authors should address why this was not done.

The graphical lines and points in Fig 5A and 5B is difficult to distinguish. The resolution of the data points should be improved. In lines 274-277 the authors states the following: "At a concentration near its MIC (2.5 mg/L), Eto alone exhibited a constant yet relatively slow bactericidal activity in vitro, resulting in an approximate 4-log₁₀ reduction of CFU in 7 days, never reaching the limit of detection of 102 CFU/mL." This is very difficult to verify. From Fig 5A it appears that 102 is indeed reached prior to regrowth, specifically at 25mg/ml and is sustained until D21. The authors should comment on this.

While, Alp/E is shown to be more effective than Eto alone, measurements terminate at 21 days. Has measurement been done for longer periods? What happens with regrowth after D21?

Why is there a limit of detection of 102 CFU's? For example, it is possible to obtain absolute pulmonary bacterial burdens (<100) if the whole lung volume is homogenised and plated. This would give an indication of whether treatment induces levels of sterility.

It appears that the addition of alpibactir to Eto beyond 15mg/ml has no added benefit in terms of reducing pulmonary bacilli burdens, at least at the 4 week time point reported in Fig 6. It may be that the optimal Eto concentration is less than 15 (between 5-15mg/ml). The authors may want to comment on this. The addition of alpibactir to Eto at a concentration at 50mg/ml has no added potentiation. The authors should comment on this observation.

Minor comments/questions:

Blue to Yellow scale need to be indicated in Fig 1A.

Line 194, should be Fig 2B-2C, rather than Fig 2A-C

Why was the concentration of 0.33mg/ml for alpibactir selected for the study in Fig 3?

the authors should provide context for the selection of concentrations used, for both Eto and alpiactir in the study section under the heading "Alpibectir/Eto inhibits growth of MDR-TB, including ethA and inhA mutated strains" page lines 304-404. What is the relevance of 1/6th of the clinical breakpoint (line 387)?

Version 1:

Reviewer comments:

Reviewer #2

(Remarks to the Author)

The manuscript is well-delineated, cogent and an exemplary illustration of the use of multidisciplinary and orthogonal approaches to tackle scientific questions. The authors addressed most comments. This reviewer has two additional, pedantic ones: the acronym DMPK, line 85, should be defined and there is a problem with color labels between the graphs and the legend in Figure 6 that must be fixed ("Alpiactir alone" is certainly wrongly labeled).

Reviewer #1 (Remarks to the Author):

The manuscript presents interesting findings on the potential of alpipectir to enhance the antibacterial activity of ethionamide (Eto) and prothionamide (Pto) through its interaction with the transcriptional regulator VirS. The study is well-designed, combining biophysical, genetic, and cellular assays to demonstrate the mechanism of action and therapeutic potential of alpipectir. However, some minor revisions are needed to strengthen the manuscript.

1. *In line 285, the author mentioned “In contrast to the intrinsic growth-inhibitory activity shown in Fig.4, no bactericidal effect was observed when alpipectir was used alone at concentrations up to 1 mg/mL” and attribute the results to the impact on bacterial adaptability. However, the animal assay of alpipectir showed a decrease of Mtb CFU, when treat with 1.6 mg/mL of alpipectir. Can the author provide a more detailed discussion on these results?*

Thank you for highlighting this important point and for noting the inconsistencies in the original paragraph. First, as shown in Fig. 6, the maximum concentration of alpipectir used in the *in vitro* time-kill assay was 0.3 mg/L and not 1 mg/L as we had previously stated. The corresponding sentence (line 287) has now been corrected to: “In comparison to the intrinsic growth-inhibitory activity shown in Fig. 5, no bactericidal activity was observed when alpipectir was used alone at concentrations up to 0.3 mg/L”.

It is important to emphasize that the checkerboard assay shown in Fig. 5 measures the evolution of bacterial density over a five-day period. In this assay, even at 1 mg/L alpipectir, the bacterial population at day 5 remains higher than at day 0 (noting that OD at day 0 is below the detection limit). The OD difference between the 0 mg/L and 1 mg/L conditions is approximately 0.08 on day 5, supporting our conclusion that alpipectir limits bacterial growth but does not exert a bactericidal effect. A similarly low, statistically non-significant inhibitory trend was observed in the kill-curve assay, which lacks the sensitivity required to detect such modest growth-inhibitory effects.

As rightly noted by the reviewer, the *in vivo* experiment shows a more pronounced growth-inhibitory effect, with a 2.05- \log_{10} CFU reduction (down to 6.84 \log_{10} CFU) at 1.6 mg/kg alpipectir compared to the untreated control on day 28 (8.89 \log_{10} CFU). This effect was not detectable at the lower tested doses (0.5 mg/kg and 0.1 mg/kg). This confirms that alpipectir, consistent with the time-kill results, can reduce bacterial multiplication *in vivo* but does not achieve bactericidal activity, which would require CFU levels to fall below 5.36 \log_{10} .

To strengthen the connection between the *in vitro* and *in vivo* observations, we have revised the corresponding sentence in the *in vivo* section (line 441) as follows: “Strikingly, alpipectir alone at 1.6 mg/kg prevented mortality ($p=0.0001$), consistent with its impact on lung CFU counts (**SI Fig.14B**) and with its intrinsic growth-inhibitory activity observed *in vitro*”.

2. *In line 258-262, the author suggested polar effect may occur during KO of adhD. The gene complementation experiment needs to be conducted.*

Response: Indeed, since we used homologous recombination to replace the *adhD* gene with a hygromycin-resistance cassette, we acknowledge the possibility of a polar effect on downstream genes. As noted in the manuscript, the loss of intrinsic activity of alpipectir observed in the $\Delta adhD$ strain may therefore not be solely attributable to the disruption of *adhD* itself.

To determine whether one or more downstream genes contribute to intrinsic activity of alpipectir, it would be necessary to generate clean, non-polar deletions of each gene in the *mymA* operon, followed by systematic complementation analyses. We have begun to initiate such an approach, and we now realize that this represents a substantial amount of work that will require many additional months.

While we agree that a detailed understanding of the contribution of each gene of the *mymA* operon is an important future objective, the focus of the present manuscript is to reveal the dual mode of action of alpipectir. We believe that an in-depth genetic dissection would refine mechanistic insight but would not alter the central conclusion regarding the involvement of the *mymA* operon.

To address the reviewer’s comment, we propose adding a clarifying sentence at line 264 indicating that identifying the specific genetic determinants underlying the intrinsic activity of alpipectir will be pursued in a future manuscript.

3. *In transcriptomic/proteomic analysis, the author only showed sharp LFC of selected genes. However, it is necessary to declare the statistical significance. Also, a volcano plot with annotation of these differentially expressed genes would also be more intuitive.*

Response: We thank the reviewer for this valuable suggestion. In response, we have repeated the transcriptomics experiment to include 3 biological replicates, which now allows us to provide statistical significance for the differentially expressed genes. We have added a volcano plot summarizing these results (new Figure 2), with annotations of key genes discussed in the manuscript.

Since the proteomics analysis (n=2) corroborates the transcriptomics data for the most highly expressed genes previously shown, we have moved the proteomics results to the Supplementary Information (SI Table 1). We also provide, in the supplementary section, the complete list of proteins detected in this experiment (Supplementary Data 1).

4. *It would be beneficial to include an INH + alpidectir treatment group as an additional control in Fig.6. This would help to distinguish whether the observed effects are unique to the Eto+alpidectir combination or not.*

Response: To address this comment, we performed an additional *in vitro* experiment as the question can be resolved without requiring a new animal study.

We evaluated the effect of alpidectir in combination with INH and observed that the addition of alpidectir did not alter the MIC of INH in a wild-type strain. This confirms that alpidectir selectively boosts Eto but not INH. The corresponding data have been included in the Supplementary Information (SI Fig. 12) and a sentence describing this result has been included in the main text (line 367).

5. *The authors presented a virS/mymA-inactivated Beijing strain in Table 3. Is this mutation frequently observed in clinical strains? This would determine the effectiveness of the combination therapy.*

Response: The mutation observed in 13MY2489 is very rare among lineage 2 strains. In the CRyPTIC database, which includes data from 12,287 *M. tuberculosis* isolates, 4,295 strains belong to lineage 2. Among these 4,295 strains, none displays a complete deletion of the *virS-mymA* regulon; only two strains contain a partial deletion of *mymA* and two additional strains contain a complete deletion of *mymA*. In contrast, as previously reported in our study on the SMART751 booster of Eto, strains belonging to sublineage 4.8 are characterized by a large genomic deletion encompassing the *virS-mymA* regulon. As a result, this sublineage does not respond to the boosting effect of alpidectir. We have added the following sentence in the corresponding paragraph (line 401) to reiterate this known limitation of the approach: "It is worth noting that *M. tuberculosis* sublineage 4.8 naturally harbors a similar large deletion encompassing the *virS-mymA* regulon³⁶, rendering these strains intrinsically unresponsive to alpidectir. Accordingly, particular caution will be necessary in regions where this sublineage is prevalent to avoid reducing the Eto dose to sub-MIC levels in patients infected with such strains."

Reviewer #2 (Remarks to the Author):

The manuscript by Edo et al. reports work using a combination of orthogonal approaches to show that Alpidectir, an activator of VirS with bactericidal activity against Mycobacterium tuberculosis (Mtb), boosts the activation of the oxidoreductase MymA, promoting ethA-independent conversion of ethionamide into its active form to potentiate inhibition of InhA. The authors show that ethionamide administered in combination with Alpidectir led to lower bacterial loads in Mtb-infected mice. The work presented in the manuscript is well executed, robust and cogent. However, some aspects of the work should be clarified and discussed in more detail. Specific comments are as follows:

1. *It is unclear whether Alpidectir alone can kill intracellular Mtb. In Figure 1F, Alpidectir at concentrations ranging from 0.050-0.333mg/L does not prevent replication of intracellular Mtb at concentrations at which ethionamide is ineffective against Mtb, suggesting that Alpidectir is not effective against intracellular Mtb. This should be clearly discussed.*

Response: We thank the reviewer for giving us the opportunity to clarify this aspect. In our *in vitro* experiments, we observed that alpipectir exhibits modest intrinsic growth-inhibitory activity when used alone but displays a strong boosting effect on the activity of ethionamide.

In the intracellular assay originally presented in Fig. 1F in the first version of the manuscript (now SI Fig. 4), differentiated THP-1 macrophages were infected with a high multiplicity of Mtb. Under these conditions, macrophage survival relies on antibiotic treatment capable of preventing infection-induced lysis. In this stringent setting, we were able to detect the boosting effect of alpipectir on ethionamide, but not its intrinsic activity. We reasoned that the intrinsic activity of alpipectir alone is insufficient to prevent macrophage lysis in this model.

To directly assess the intrinsic intracellular activity of alpipectir, we have performed an additional assay in which THP-1 cells were simultaneously differentiated and infected with luciferase-expressing Mtb, followed by 5 days of incubation with antibiotics. In this setup, macrophages remain viable regardless of treatment (new figure added in Supplementary Information, SI Fig. 3), allowing direct quantification of intracellular bacterial luminescence. This new experiment (new Fig. 1F) shows both the intrinsic activity of alpipectir and its boosting effect on ethionamide. Importantly, the magnitude of the boosting effect (ca. 10-fold decrease of the IC₉₀ of ethionamide in the presence of alpipectir) is consistent across both intracellular assays.

Accordingly, we have moved the initial experiment to the Supplementary Information (SI Fig. 4) and added a clarifying sentence in the main text (line 134) to reflect these updates.

- 2. In addition, the authors use a rather unusual murine model of tuberculosis wherein the drug is administered 24 hours post infection. Rationale for using this model should be included. Did the lack of effect of Alpipectir against intracellular Mtb motivate the use of this model?. Does the short time lapse between infection and treatment amplify the effect of Alpipectir alone because the drug helps reduce the extracellular Mtb load?*

Response: We used the preventive murine model, in which treatment is initiated one day after infection, when the bacterial load remains limited (10⁵ to 10⁶ CFU). The model can be applied to BALB/c or C57BL/6 mice, with a treatment duration of 4 weeks. This model is particularly suited for comparing the activity of different compounds used alone, which was the rationale for its use in our study. Importantly, it parallels human early bactericidal activity (EBA) studies by reproducing the early, rapidly replicating phase of infection and allowing the measurement of short-term reductions in bacterial burden. Our team, along with others, including the expert group of Eric Nuermberger, uses this model for the preclinical evaluation of novel antitubercular agents. We have added the supplementary reference below to our text (line 411) to reflect this.

Nuermberger EL et al. GSK2556286 Is a Novel Antitubercular Drug Candidate Effective In Vivo with the Potential To Shorten Tuberculosis Treatment. *Antimicrob Agents Chemother.* 2022 Jun 21;66(6):e0013222. doi: 10.1128/aac.00132-22.

- 3. Figure 2, panel B. Although the data are compelling, some relevant controls are missing in the thermal denaturation experiment. The authors should either repeat it with inclusion of another protein that does not bind Alpipectir, or ideally with inclusion of a variant of VirS that cannot bind Alpipectir. The structural data should help in designing such a variant.*

Response: We thank the reviewer for this valuable suggestion. In accordance with the comment, we have included α -lactalbumin as a negative control in the thermal denaturation experiment. No effect of alpipectir on the thermal denaturation profile of α -lactalbumin was observed, indicating the absence of binding. This additional control experiment has been included in the Supplementary Information (SI Fig. 6) and mentioned in the main text (line 206).

- 4. Figure 4. There is no attempt to complement the $\Delta adhD$ mutant. It would be useful to complement the $\Delta adhD$ mutant with *adhD* and each of the three downstream genes individually first, and then in combination if unsuccessful, to define which gene(s) are responsible for the intrinsic activity of Alpipectir.*

Response: We thank the reviewer for this comment. Please refer to our response to Comment 2 from Reviewer 1, which addresses this point in detail.

- 5. Figure 5. The molecular mechanisms underlying ethionamide potentiation by Alpipectir remain unclear. Ethionamide is primarily converted into its active form by EthA in the absence of*

Alpibectir, the addition of which favours bioconversion of ethionamide by MymA. If EthA and MymA activate ethionamide into the same bioactive form (the NAD adduct), do they do so at the same rate? For the same concentration of ethionamide, how do levels of the bioconverted form compare in H37Rv versus H37Rv treated with Alpibectir? Does activation by MymA allow for larger accumulation of the bioconverted, active form compared to activation by EthA? This could explain why the combination of Alpibectir/ethionamide is better able to kill drug-recalcitrant Mtb subpopulations, and why the combination is effective at killing strains overexpressing InhA (lines 368-373). These measurements should be done; at the very least, these questions should be discussed.

We thank the reviewer for these insightful questions. The mechanisms underlying ethionamide activation by EthA and MymA are not completely understood despite substantial efforts from several groups. To date, the relative rates of ethionamide activation by EthA and MymA have not been determined and therefore cannot be directly compared. Measuring the rate of ethionamide bioconversion has proven technically challenging for many groups, including ours, as the pathway proceeds through the ethionamide S-oxide (Eto-SO) intermediate, likely involves radical species, and ultimately results in the formation of the ethionamide-NAD adduct (see Johnston *et al.* 1967). Importantly, this Eto-NAD adduct has never been directly quantified in *M. tuberculosis*; its existence is demonstrated solely through crystallographic structure of InhA in complex with this adduct. Consequently, to the best of our knowledge, no reliable method currently exists to precisely quantify either the rate of ethionamide activation or the intracellular accumulation of its bioactive species.

What we have demonstrated, however, is that alpibectir-induced overexpression of *mymA* (as evidenced by our transcriptomic and proteomic analyses) confers ethionamide hypersensitivity with MIC shifts equivalent to those observed upon *ethA* de-repression by EthR inhibitors (BDM31343, BDM41906; see Willand *et al.* 2009 and Flipo *et al.* 2011).

This functional equivalence strongly suggests that both conditions achieve comparable steady-state levels of the bioactive Eto-NAD species, despite utilizing different monooxygenases.

Moreover, overexpression of *inhA* reduces the efficacy of both alpibectir (SI Fig. 11) and BDM41906 to comparable extents, confirming that: (i) both monooxygenases generate the same InhA-targeting active species (Eto-NAD) and (ii) the boosted monooxygenase activities under alpibectir (MymA-mediated) and BDM41906 (EthA-mediated) treatment are functionally equivalent in terms of ethionamide bioactivation capacity.

Per the reviewer's request, we have now discussed this aspect at line 496 of the main text.

References:

- Johnston JP *et al.* The metabolism of ethionamide and its sulfoxide. J Pharm Pharmacol. 1967 Jan;19(1):1-9. doi: 10.1111/j.2042-7158.1967.tb07986.x.
- Willand N *et al.* Synthetic EthR inhibitors boost antituberculous activity of ethionamide. Nat Med. 2009 May;15(5):537-44. doi: 10.1038/nm.1950.
- Flipo M *et al.* Ethionamide boosters. 2. Combining bioisosteric replacement and structure-based drug design to solve pharmacokinetic issues in a series of potent 1,2,4-oxadiazole EthR inhibitors. J Med Chem. 2012 Jan 12;55(1):68-83. doi: 10.1021/jm200825u.
- DeBarber AE *et al.* Ethionamide activation and sensitivity in multidrug-resistant Mycobacterium tuberculosis. Proc Natl Acad Sci U S A. 2000 Aug 15;97(17):9677-82. doi: 10.1073/pnas.97.17.9677.
- Wang F *et al.* Mechanism of thioamide drug action against tuberculosis and leprosy. J Exp Med 2007 Jan 22;204(1):73-8. doi: 10.1084/jem.20062100.
- Ushtanit A *et al.* Molecular Determinants of Ethionamide Resistance in Mycobacterium tuberculosis. Int J Mol Sci 2022 Jan 20;11(2):133. doi: 10.3390/antibiotics11020133.
- Laborde J *et al.* Ethionamide biomimetic activation and an unprecedented mechanism for its conversion into active and non-active metabolites. Org Biomol Chem 2016 Sep 21;14(37):8848-8858. doi: 10.1039/c6ob01561a.

6. - *p12, line 282: the use of "only" might be misleading as ethionamide alone at 2.5 or 25 mg/L leads to statistically similar levels of bacterial clearance at day 7.*

Response: The reviewer is correct. We have removed the word "only" from line 284 (originally line 282).

7. - *The grounds on which the authors conclude that InhA is expressed at a basal level or overexpressed are unclear.*

Response: We thank the reviewer for highlighting this important point. To provide evidence showing overexpression of *inhA*, we have now included additional experimental validation. We repeated our experiments using a well-characterized and previously published *inhA* overexpressing strain (see reference below). This strain exhibits a 30-fold increase in INH MIC compared to H37Rv, confirming functional *inhA* overexpression as previously described.

We have updated SI Fig. 11 in the Supplementary Information accordingly.

Hartkoorn RC, Sala C, Neres J, Pojer F, Magnet S, Mukherjee R, Uplekar S, Boy-Röttger S, Altmann KH, Cole ST. Towards a new tuberculosis drug: pyridomycin - nature's isoniazid. *EMBO Mol Med*. 2012 Oct;4(10):1032-42. doi: 10.1002/emmm.201201689

8. - How do the authors explain the absence of dose effect between administration of 15 and 50mg/kg of ethionamide?

Response: We thank the reviewer for inviting us to clarify this observation. When administered as a monotherapy, ethionamide displays a clear dose-dependent bactericidal effect between 5 and 50 mg/kg (exceeding a 3-log₁₀ reduction), followed by a plateau corresponding to target saturation between 50 and 200 mg/kg. In the presence of alpipectir (0.5 or 1.6 mg/kg), the ethionamide dose-response curve is shifted leftward, with the maximal antibacterial effect, slightly surpassing that of 200 mg/kg ethionamide alone, achieved at 15 mg/kg. These findings indicate that the pharmacodynamic saturation point (InhA inhibition), which is only reached at ≥50 mg/kg under ethionamide monotherapy, is attained at 15 mg/kg when alpipectir is co-administered. Consequently, increasing the ethionamide dose to 50 mg/kg in the presence of alpipectir does not confer any additional efficacy, as the pharmacological target is already saturated at 15 mg/kg.

This clarification has been added to the revised manuscript at line 429.

9. - Please check the policy of Nature Communications regarding the use of “data not shown”.

Response: We initially used “data not shown” on 2 instances.

For the first instance (originally line 186; now line 193), we have replaced the phrase by adding the corresponding data as a new panel in the Supplementary Information (SI Fig. 5) which shows the synthetic gene circuit experiments assessing the interaction of EthR2 and EthR with alpipectir.

For the second instance (originally line 286; now line 287), we have modified our sentence to only mention the results that are shown in the survival kinetics analysis (Fig. 6).

Reviewer #3 (Remarks to the Author):

The objective of discovering new treatments for tuberculosis is an undeniable priority, especially for those areas and regions and which tuberculosis have a high incidence for example in sub-Saharan Africa and in south-east Asia. The reporting of novel drugs or drug combinations are therefore not only important but necessary.

In this study Edoo and colleagues report on the novel drug combination that involves Alpipectir and ethionamide. This study reports on it to effects in studies that were directly conducted against M. tuberculosis bacilli, against bacilli harboured in macrophages as well as its effects tested in mice.

General Comments:

The study design is well constructed with logical flow in the experimental steps. The findings that alpipectir potentiates Eto activity and has antibacterial activity independent of Eto is highly promising.

- 1. In Fig.1A the effects of alpipectir is generally evaluated against the effect of Eto in its absence. While this is a necessary control, the inclusion of a compound in the same class of amino piperidines that does not potentiate Eto is an additional control required to demonstrate specificity.*

Response: We thank the reviewer for this comment. As requested by the reviewer, we performed an additional experiment to evaluate the effect of a close analogue of alpipectir on the growth inhibitory activity of ethionamide. The hydrophobic electron-withdrawing trifluoromethyl group was replaced by a more polar, electron-donating group (aminomethyl) on the spiroisoxazoline motif. With this analogue, we observed no boosting effect on ethionamide activity, nor intrinsic antibacterial activity, clearly

demonstrating that it does not bind VirS and does not induce expression of the *mymA* operon. These results are fully consistent with the VirS Alpibectir X-ray co-structure, in which the trifluoromethyl group engages hydrophobic contacts with Phe197 and Ile101, interactions that the aminomethyl group is unlikely to support. In addition, the bend imposed by the nitrogen atom may introduce steric clashes that further prevent binding. We have added a figure in the Supplementary Information (SI Fig. 2), as well as a corresponding sentence in the main text at line 121.

- In the next phase of the study, the effects of Alp/E was tested against intracellular bacilli (Fig 1F). The authors report efficacy, however whether macrophages remain intact and viable for the duration of the assay is not presented. Should macrophages be lysed, the test would essentially evaluate efficacy against extracellular bacilli. The inclusion of a data that demonstrate macrophage viability for the duration of the experiment must be included to validate the conclusion of efficacy against intracellular bacilli. Again, while the absence of alpibactir serves as a control in Fig 1F, a control to demonstrate drug specificity is absent.*

Response: In the intracellular assay originally presented in Fig.1F (now moved to the Supplementary Information as SI Fig. 4), THP-1 cells were first differentiated and then infected with *M. tuberculosis* at a multiplicity of infection of 10, a condition that leads to macrophage lysis in the absence of antibiotic treatment, as evidenced by increased resazurin fluorescence in wells containing the lowest antibiotic concentrations. This experiment enabled us to determine the concentration of AlpE required to prevent macrophage lysis caused by *M. tuberculosis* infection.

In response to comment 1 from Reviewer 2, we have performed an additional intracellular assay in which THP-1 cells were simultaneously differentiated and infected with luciferase-expressing Mtb. In this new setup, we have used THP1-cells stably expressing a red fluorescence protein, enabling monitoring of macrophage viability over the 5-day infection period (See the following references describing the model). Macrophages remained viable across all treatment groups. These new data have been added to the Supplementary Information (SI Fig. 3), and a clarifying sentence has been incorporated in the main text (line 134) to reflect these updates.

References:

- Zheng X, Av-Gay Y. 2017. System for Efficacy and Cytotoxicity Screening of Inhibitors Targeting Intracellular *Mycobacterium tuberculosis*. J. Vis. Exp., e55273, doi: 10.3791/55273.
- Sorrentino F, Gonzalez del Rio R, Zheng X, Presa Matilla J, Torres Gomez P, Martinez Hoyos M, Perez Harran ME, Mendoza Losana A, Av-Gay Y. 2016. Development of an intracellular screen for new compounds able to inhibit *Mycobacterium tuberculosis* growth in human macrophages. Antimicrob Agents Chemother 60:640–645. doi: 10.1128/AAC.01920-15.
- Kavanagh ME, McLean KJ, Gilbert SH, Amadi CN, Snee M, Tunnicliffe RB, Arora K, Boshoff HIM, Fanourakis A, Rebollo-Lopez MJ, Ortega F, Levy CW, Munro AW, Leys D, Abell C, Coyne AG. 2025. J Med Chem 68 (14), 14416-14441. doi: 10.1021/acs.jmedchem.5c00478

- In line 169, the authors refer to the "...exceptional specificity of alpibactir...". The inclusion of a non-specific control for alpibactir would enhance this claim. The authors should address why this was not done.*

Response: We apologize for the confusing sentence. By using the word "specificity" in line 169 (now line 177), we were not referencing the structural determinants of alpibectir but rather the functional selectivity of alpibectir towards the *mymA* operon and to a lesser extent, towards EthA2. We have replaced "specificity" in line 177 by "functional selectivity" to make the context explicit.

- The graphical lines and points in Fig 5A and 5B is difficult to distinguish. The resolution of the data points should be improved. In lines 274-277 the authors states the following: "At a concentration near its MIC (2.5 mg/L), Eto alone exhibited a constant yet relatively slow bactericidal activity in vitro, resulting in an approximate 4-log10 reduction of CFU in 7 days, never reaching the limit of detection of 102 CFU/mL." This is very difficult to verify. From Fig 5A it appears that 102 is indeed reached prior to regrowth, specifically at 25mg/ml and is sustained until D21. The authors should comment on this.*

Response: We agree that the original presentation of Fig. 5A and 5B (now Fig. 6A and 6B) made the data difficult to interpret. Two issues contributed to this:

1. the near-complete overlap of five series of data points at day 7 and
2. the difficulty distinguishing cultures that exhibited regrowth at days 14 and 21 from those that did not. To address this, we have separated the data of each strain into two independent graphs, one showing cultures treated with compounds alone and the other showing cultures treated with Eto + alpipectir. This presentation clearly highlights the absence of bacterial rebound in the presence of alpipectir, in contrast to the conditions where Eto is used alone.

Regarding the reviewer's specific point on the time-kill curves with Eto monotherapy: at 2.5 mg/L, the limit of detection was not reached at day 7; the data point lies slightly above the 10^2 CFU/mL threshold (please refer to the source data of Fig. 6 for the exact values). In contrast, for Eto at 25 mg/L, the limit of detection was indeed reached at day 14. However, the regrowth observed at day 21 demonstrates that reaching the detection limit does not equate complete sterilization under these conditions.

5. *While, Alp/E is shown to be more effective than Eto alone, measurements terminate at 21 days. Has measurement been done for longer periods? What happens with regrowth after D21?*

Response: No measurements were done beyond 21 days; therefore, we do not claim complete sterilization of the cultures. However, the rebound observed at day 21 in the condition where bacteria were undetectable at 14 days (Eto 25 mg/L) indicates that, if surviving bacilli remain, they are present at extremely low concentration. While their persistence cannot be fully excluded, this observation must be interpreted in the context of clinical polytherapeutic regimens. In patients, AlpE would not constitute the sole active antimicrobial component, and any rare surviving bacilli would be expected to be eliminated by the additional drugs present in the treatment combination.

6. *Why is there a limit of detection of 10² CFU's? For example, it is possible to obtain absolute pulmonary bacterial burdens (<100) if the whole lung volume is homogenised and plated. This would give an indication of whether treatment induces levels of sterility.*

Response: The limit of detection in the *in vivo* experiment was $1.24 \log_{10}$ CFU, and this has now been added to Fig. 7 for clarity. Importantly, none of the lung samples reached this detection threshold. If the reviewer is instead referring to the *in vitro* time-kill curves, the LOD is determined by the experimental setup: 10 μ L of culture were plated at each time point, corresponding to a detection limit of $2 \log_{10}$ CFU/mL. By definition, time-kill assays involve sampling and plating only a small, fixed volume of the culture at each time point, which intrinsically imposes a higher LOD than would be achieved by plating the entire culture volume.

7. *It appears that the addition of alpipectir to Eto beyond 15mg/ml has no added benefit in terms of reducing pulmonary bacilli burdens, at least at the 4 week time point reported in Fig 6. It may be that the optimal Eto concentration is less than 15 (between 5-15mg/ml). The authors may want to comment on this. The addition of alpipectir to Eto at a concentration at 50mg/ml has no added potentiation. The authors should comment on this observation.*

Response to first part of comment: The reviewer is correct. The opportunity offered by alpipectir lies in achieving ethionamide maximal activity (*i.e.* E_{max}) with lower ethionamide doses. We agree that the optimal ethionamide dose lies between 5 mg/kg (which, even in combination with alpipectir, does not reach E_{max}) and 15 mg/kg (which achieves E_{max} when combined with either 0.5 or 1.6 mg/kg of alpipectir). We added a sentence to mention this at line 429.

Response to second part of comment: Please see below our response to reviewer 2 who basically raised the same comment.

Reviewer #2 : How do the authors explain the absence of dose effect between administration of 15 and 50mg/kg of ethionamide?

Response: We thank the reviewer for inviting us to clarify this observation. When administered as a monotherapy, ethionamide displays a clear dose-dependent bactericidal effect between 5 and 50 mg/kg (exceeding a 3-log_{10} reduction), followed by a plateau corresponding to target saturation between 50 and 200 mg/kg. In the presence of alpipectir (0.5 or 1.6 mg/kg), the ethionamide dose-response curve is shifted leftward, with the maximal antibacterial effect, slightly surpassing that of 200 mg/kg ethionamide alone, achieved at 15 mg/kg. These findings indicate that the pharmacodynamic saturation point (InhA inhibition), which is only reached at ≥ 50 mg/kg under ethionamide monotherapy, is attained at 15 mg/kg when alpipectir is co-administered. Consequently, increasing the ethionamide dose to 50

mg/kg in the presence of albipectir does not confer any additional efficacy, as the pharmacological target is already saturated at 15 mg/kg. This clarification has been added to the revised manuscript at line 429.

8. *Minor comments/questions:*

Blue to Yellow scale need to be indicated in Fig 1A.

Response: We have modified Fig. 1A accordingly.

9. *Line 194, should be Fig 2B-2C, rather than Fig 2A-C*

Response: We have corrected this oversight.

10. *Why was the concentration of 0.33mg/ml for albipactir selected for the study in Fig 3?*

Response: With 0.33 mg/L of albipectir, we are able to obtain maximum boosting effect of ethionamide and maximum intrinsic activity of albipectir (as shown in Fig. 1C and 1D). This has prompted us to use this concentration of albipectir for several experiments.

The authors should provide context for the selection of concentrations used, for both Eto and albipactir in the study section under the heading "Albipactir/Eto inhibits growth of MDR-TB, including ethA and inhA mutated strains" page lines 304-404. What is the relevance of 1/6th of the clinical breakpoint (line387)?

Response: We thank the reviewer for pointing out the absence of an explanation for the concentrations we used in this experiment. The ethionamide concentration used in the MGIT assay was set at one sixth of the clinical breakpoint, a sub-inhibitory level chosen to permit detection of sensitization effects. At this concentration, growth of *M. tuberculosis* is only partially inhibited, allowing quantification of changes in drug susceptibility by albipectir. We selected two concentrations of albipectir to capture both the plateau and the ascending phase of the potentiation curve of albipectir. The maximal concentration (0.05 mg/L) confirms the full extent of the effect, while the lower concentration (0.017 mg/L) demonstrates that the response is dose-dependent and not a non-specific consequence of high compound exposure. We have added a sentence to explain the concentrations chosen at line 384.

Reponse to Reviewers' Comments

Reviewer #2 (Remarks to the Author):

The manuscript is well-delineated, cogent and an exemplary illustration of the use of multidisciplinary and orthogonal approaches to tackle scientific questions. The authors addressed most comments.

This reviewer has two additional, pedantic ones: the acronym DMPK, line 85, should be defined and there is a problem with color labels between the graphs and the legend in Figure 6 that must be fixed ("Alpibactir alone" is certainly wrongly labeled).

We thank the reviewer for their thoughtful comment and for pointing out these oversights. We have duly amended the manuscript.